# CORTICAL POLICY: A DUAL-STREAM VIEW TRANSFORMER FOR ROBOTIC MANIPULATION

**Xuening Zhang**[1]**, Qi Lv**[1]**, Xiang Deng**[†1]**, Miao Zhang**[1]**, Xingbo Liu**[2]**, Liqiang Nie**[1]
[1]Harbin Institute of Technology (Shenzhen), Shenzhen, Guangdong 518055, China
[2]Shandong Jianzhu University, Jinan, Shandong 250101, China

## ABSTRACT

View transformers process multi-view observations to predict actions and have shown impressive performance in robotic manipulation. Existing methods typically extract static visual representations in a view-specific manner, leading to inadequate 3D spatial reasoning ability and a lack of dynamic adaptation. Taking inspiration from how the human brain integrates static and dynamic views to address these challenges, we propose **Cortical Policy**, a novel dual-stream view transformer for robotic manipulation that jointly reasons from static-view and dynamic-view streams. The static-view stream enhances spatial understanding by aligning features of geometrically consistent keypoints extracted from a pretrained 3D foundation model. The dynamic-view stream achieves adaptive adjustment through position-aware pretraining of an egocentric gaze estimation model, computationally replicating the human cortical dorsal pathway. Subsequently, the complementary view representations of both streams are integrated to determine the final actions, enabling the model to handle spatially-complex and dynamically-changing tasks under language conditions. Empirical evaluations on RLBench, the challenging COLOSSEUM benchmark, and real-world tasks demonstrate that Cortical Policy outperforms state-of-the-art baselines substantially, validating the superiority of dual-stream design for visuomotor control. Our cortex-inspired framework offers a fresh perspective for robotic manipulation and holds potential for broader application in vision-based robot control.

## 1 INTRODUCTION

Enabling robots to handle the uncertainty of unstructured, non-stationary environments remains a fundamental challenge for robotic manipulation (Liang et al., 2024; Chu et al., 2025; Li et al., 2025a). Critically, this challenge requires coherent scene perception through robust fusion of multimodal inputs, including vision, language, and proprioception (Lv et al., 2024). To achieve this, view transformers (Goyal et al., 2023; 2024) provide an efficient solution by leveraging multi-view images to predict actions, showing competitive performance while offering greater scalability than explicit 3D representation-based approaches.

Leveraging a set of static camera views around the robot workspace, previous view transformers typically extract visual representations through naively fusing view-specific 2D information. Despite demonstrated competence in stationary environments (Goyal et al., 2023; 2024; Zhang et al., 2024; Qian et al., 2025), this paradigm fails to model cross-view relationships, hampering 3D spatial understanding beyond 2D images. More importantly, the static camera configurations lack dynamic-view perception essential for human-like manipulation with unpredictable object displacements (Hallquist et al., 2024). These limitations manifest as two frequent failure modes in robotic manipulation: **inadequate spatial reasoning** and **dynamic adaptation failure**. As shown in Fig. 1 (top), when placing an object in between the others, the SOTA view transformer (Goyal et al., 2024) exhibits significant error, failing to place within the right region. This lack of robustness to 3D scene structure is further supported by findings that view transformers are sensitive to environmental disturbances like texture, lighting, and table color variations (Qian et al., 2025). Furthermore, when the target object is moved during approach, existing methods (Goyal et al., 2023; 2024; Qian et al., 2025) fail

---

[†]Corresponding author: dengxiang@hit.edu.cn

to adjust trajectories as expected, persisting with the originally-planned trajectory until task failure (Fig. 1, bottom). These empirical findings underscore that the scene perception provided by current view transformers remains incomplete, which hinders robust manipulation performance.

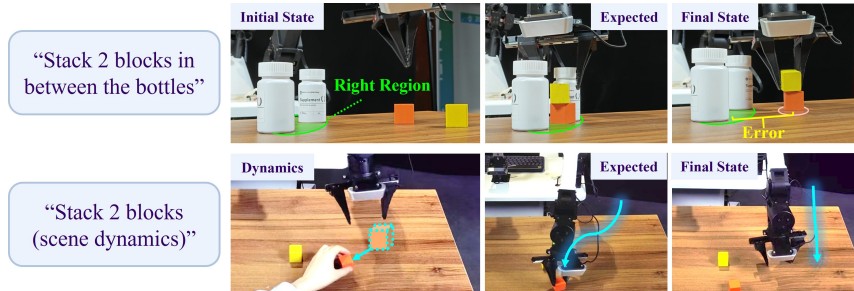

Figure 1: **Deficiencies of prior view transformers for robotic manipulation.** (Top) A task requiring the robot to understand the spatial relationships between two bottles before deciding on the placing position. Previous method RVT-2 fails to merge different camera views correctly in 3D, causing wrong block placement. (Bottom) Dynamic adaptation failure during object displacement.

To bridge this gap, we draw inspiration from how the human brain organizes view-based visual cues to guide behavior, and in particular, two cortical pathways for visual processing. The ventral pathway utilizes static views for scene understanding, while the dorsal pathway specializes in dynamic-view perception, leveraging real-time visual feedback to adjust trajectory (Rossit & McIntosh, 2021; Chen et al., 2025). Translating this cortical principle into a computational framework, we propose Cortical Policy, a dual-stream view transformer for endowing robots with integrated 3D spatial understanding and dynamic adaptation. Our method enhances robotic perception through two separate, complementary pipelines: a static-view stream that encodes enduring environmental structures and a dynamic-view stream that derives actions from motion cues.

To enhance spatial comprehension, the static-view stream learns 3D-aware features by enforcing cross-view geometric consistency, which is supervised by a 3D foundation model. To facilitate adaptive re-planning under task dynamics, the dynamic-view stream extracts action-oriented features and heatmaps from a pretrained, position-aware transformer adapted from an egocentric gaze estimation model. By integrating view representations of both streams, Cortical Policy generates actions that are simultaneously geometrically grounded and dynamically adaptive. Extensive evaluations on RLBench, COLOSSEUM, and real-world tasks demonstrate that our cortex-inspired policy, with enhanced 3D awareness and adaptive motion control, substantially outperforms state-of-the-art baselines. Specifically, it exhibits superior robustness against environmental perturbations and boosts the interactive capabilities of an embodied agent in dynamic physical environments. The main contributions of this work are summarized as follows:

- Different from prior view transformers that perform single-stream processing on static views, we propose Cortical Policy, a dual-stream view transformer that integrates static and dynamic views for robotic manipulation, mirroring the two human cortical pathways to advance visuomotor imitation learning.
- Unlike view-independent processing in prior methods, we introduce a cross-view geometric consistency learning objective. This objective leverages a pretrained 3D foundation model (VGGT) to align cross-view features in a shared 3D space, significantly enhancing the spatial reasoning robustness of the static-view stream.
- A novel dynamic-view stream absent in prior work is designed to emulate the human dorsal pathway. This stream extracts action-oriented representations from a position-aware, pretrained gaze estimation model, thereby enabling adaptive trajectory adjustment.

## 2 RELATED WORK

This work extends view transformers for robotic manipulation by enhancing static-view 3D perception and introducing dynamic-view processing. We review the relevant work in this section.

**View Transformers for Robotic Manipulation.** View transformers have become a prevalent architecture for language-conditioned manipulation (Guhur et al., 2022; Ma et al., 2024). They aggregate multi-view visual inputs with language instructions and proprioception to predict 6-DoF gripper poses, states, and collision indicators. RVT (Goyal et al., 2023) establishes a five-camera paradigm (back, front, top, left, right) to render virtual static views, using a view transformer to predict view-specific heatmaps, which are back-projected to 3D to estimate gripper translation; multi-camera features are concatenated to predict the remaining action components. To improve precision, VIHE (Wang et al., 2024) and RVT-2 (Goyal et al., 2024) adopt multi-stage refinement: VIHE iteratively renders virtual in-hand static views, while RVT-2 localizes regions of interest with three static views (front, top, right) before predicting poses from refined regions. Recent methods enhance static-view visual representations with visual foundation models (Zhang et al., 2024; Fang et al., 2025) or 3D multi-view pretraining (Qian et al., 2025). Although these methods have advanced static-view perception, their inherent reliance on pre-defined viewpoints limits the adaptability in dynamic scenarios. In contrast, Cortical Policy jointly leverages static and dynamic views for action prediction to overcome this limitation.

**3D Perception in Robotics.** To enhance robots' understanding of the physical world, extensive efforts have been made to integrate 3D representations into robotic manipulation (James et al., 2022; Goyal et al., 2023; Lv et al., 2025). Existing approaches, however, face distinct challenges. Voxel-based methods (James et al., 2022; Shridhar et al., 2023) are computationally expensive. Point cloud methods handle occlusion and sim-to-real transfer well, yet require fine-grained semantic alignment (Zhen et al., 2024; Cui et al., 2025) or use inefficient backbones (Chen et al., 2024). Multi-perspective projection offers an efficient alternative by projecting point clouds onto virtual orthographic views to generate multi-camera RGB-D images, and has been widely adopted in recent work (Goyal et al., 2023; 2024; Fang et al., 2025; Li et al., 2025b). Unlike existing multi-perspective policies that struggle to capture cross-view relations, Cortical Policy addresses this limitation by explicitly modeling inter-view relationships, enhanced by geometric priors from VGGT (Wang et al., 2025), a powerful 3D foundation model whose spatial knowledge remains novel in robotic manipulation (Lin et al., 2025; Tang et al., 2025; Abouzeid et al., 2025). We introduce a novel integration of VGGT within the static-view stream, using its predictions to enforce view-invariant feature learning.

## 3 METHOD

### 3.1 PRELIMINARIES

Research on the human visual system and neuroscience reveals several cortical principles, which could guide the development of manipulation policies and enable robots to achieve human-like proficiency. These principles include:

1. **Parallel streams with separable and complementary structures and functions.** The dorsal and ventral streams emerge from distinct regions of the early visual cortex, processing dynamic and static visual signals respectively (Chen et al., 2025). Separate processing channels support generalization to novel scenes and adaptability to dynamic tasks.
2. **Dual-stream visuomotor control**. Consistent with visual processing, human visuomotor control follows a dual-stream pattern (Rossit & McIntosh, 2021): the ventral stream handles scene perception and object identification, while the dorsal stream translates retinal input into adaptive motor signals. Both streams are indispensable for precise motor control.
3. **Enduring representations in the ventral stream**. The ventral stream encodes stable visual stimuli for cognitive processes (Kravitz et al., 2011; Becker et al., 2025). Using an allocentric (world-centered) frame of reference (Milner & Goodale, 2008), it forms enduring representations that facilitate recognition, long-term memory, and action planning.
4. **Adaptive action reasoning in the dorsal stream**. The dorsal stream encodes spatiotemporal dynamics to guide actions (Kravitz et al., 2013; Hallquist et al., 2024). Using an egocentric (body-centered) frame of reference (Gheihman et al., 2025), it estimates properties of the target object in real time and adjusts movement trajectories accordingly.

Building on these cortical principles of visuomotor control, we present CORTICAL POLICY, an imitation learning framework for robotic manipulation. As illustrated in Fig. 2, the proposed policy centers on a dual-stream view transformer that integrates parallel streams: (i) a *static-view stream* encodes 3D spatial structures of the task scene through geometrically consistent representation learn-

ing, which is supervised by a pretrained 3D reconstruction model, *i.e.*, VGGT; (ii) a *dynamic-view stream* predicts adaptive actions through a position-aware pretrained model. This model processes dynamic wrist-view frames to estimate end-effector locations, generating action-oriented features that facilitate overall visuomotor reasoning. The complementary representations from both streams are fused by an action head, generating precise actions for robot control.

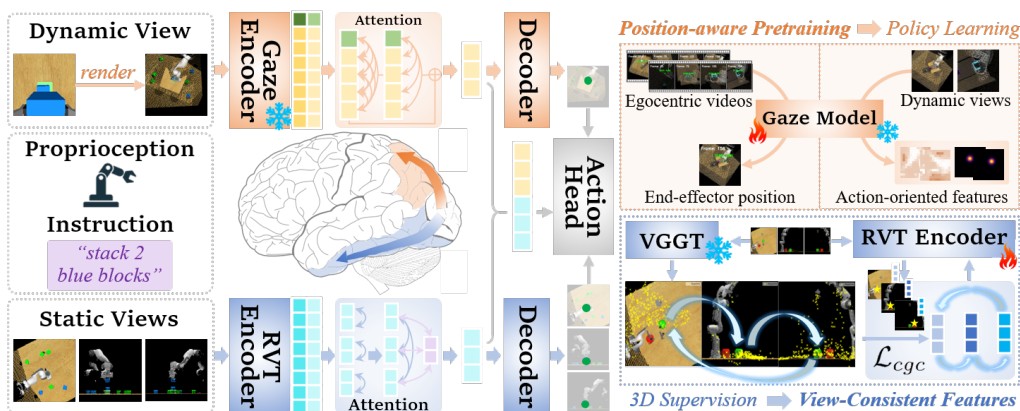

Figure 2: **Overview of the proposed cortical policy.** Inspired by the dorsal-ventral pathways in visual neuroscience, this architecture implements dual processing streams: a static-view stream for 3D spatial understanding and a dynamic-view stream for end-effector position awareness.

## 3.2 STATIC-VIEW STREAM

An enduring visual representation in the brain generally exhibits a unified and compact understanding of the 3D world, allowing easy generalization across environments, objects, and time. However, most off-the-shelf vision encoders fall short of comprehensive 3D understanding as they are trained solely on 2D images. To extract 3D-aware representations from image inputs, additional priors must be injected, typically via depth modality integration (Wu et al., 2025), cross-view consistency (You et al., 2025), or equivariance constraints (Howell et al., 2023). Notably, You et al. (2025) and Lee et al. (2025) demonstrated that incorporating view equivariance into 2D foundation models significantly boosts 3D task performance.

Motivated by these findings, our static-view stream reinforces cross-view feature consistency to learn 3D-aware semantic representations. We adapt RVT-2 backbone (Goyal et al., 2024), preserving its core mechanisms including two-stage processing, intra-view self-attention and vision-language co-attention, while augmenting its feature extractor (RVT Encoder) with geometric constraints.

**3D Supervision Generation.** We leverage *geometrically consistent keypoints* as 3D supervision signals, which represent identical 3D points across viewpoints. This design anchors cross-view consistency directly in 3D geometry. Using spatial reasoning capabilities of VGGT, we predict depth map, confidence map, camera parameters for $N$ static-view images. These predictions enable unprojection to camera-coordinate point maps $\{P_i\}_{i=1}^{N}$, which are then transformed into the world coordinate system to identify co-visible 3D points. We apply non-maximum suppression to the first viewpoint's co-visible points, selecting the $M$ highest-confidence points as candidate keypoint set $\mathcal{K}_1$. These candidates are tracked across viewpoints to yield geometrically consistent keypoint sets $\{\mathcal{K}_i\}_{i=1}^{N}$. As shown in Fig. 3 (a), VGGT-derived keypoints primarily lie on the surfaces of objects or robots, providing geometric cues for task-relevant 3D structures.

**Feature Consistency Optimization.** Given geometrically consistent keypoints $\mathcal{K} = \{(\mathbf{k}_i^{v_j})_{j=1}^{N}\}_{i=1}^{M}$, where keypoint $\mathbf{k}_i^{v_j}$ represents 3D point $\mathbf{p}_i$ in viewpoint $v_j$, we supervise RVT Encoder to align cross-view features at these keypoints. To enable fine-grained 3D supervision, we incorporate a trainable $3\times3$ convolutional layer after the RVT Encoder, which enhances feature resolution through local patch interactions (You et al., 2025). For each keypoint $\mathbf{k}_i^{v_j}$, we extract its feature $\mathbf{f}_i^{v_j}$ via bilinear sampling from the view feature map. The training objective adopts SmoothAP loss (Brown et al., 2020), which optimizes cross-view feature rankings by prioritizing similarity for geometrically consistent keypoints. For query $\mathbf{k}_i^{v_p}$ and target viewpoint $v_q$, the positive and negative

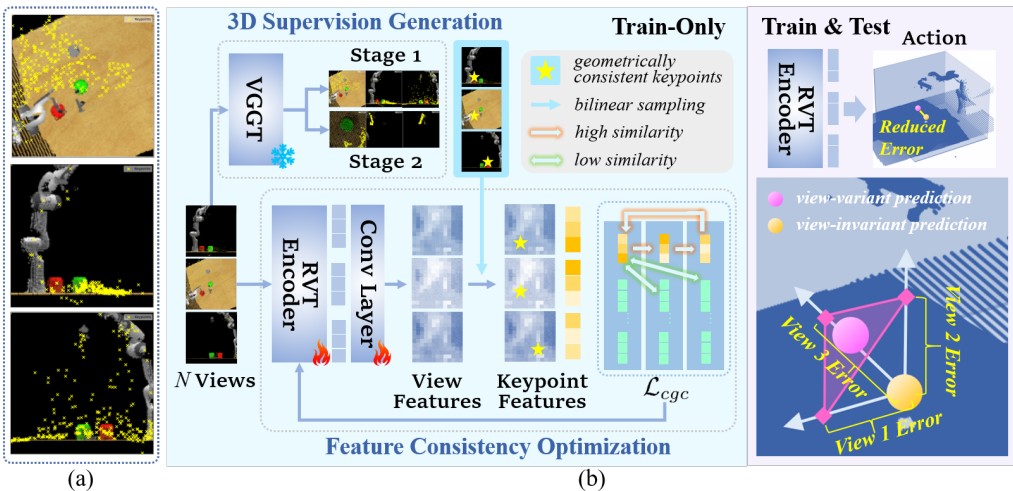

Figure 3: **Static-view stream.** (a) Visualization of geometrically consistent keypoints. (b) Pipeline.

sets are defined as $\mathcal{K}(i) = \{\mathbf{k}_i^{v_q}\}$ and $\mathcal{N}(i) = \{\mathbf{k}_j^{v_q} \mid j \neq i, \|\mathbf{p}_i - \mathbf{p}_j\|_2 > \zeta\}$ respectively, where $\zeta$ is a tunable 3D distance threshold. The SmoothAP loss enforces ranking $\mathcal{K}(i)$ above $\mathcal{N}(i)$:

$$\text{SmoothAP}(v_p \to v_q) = \frac{1}{|\mathcal{K}_p|} \sum_{i=1}^{|\mathcal{K}_p|} \frac{1 + \sum_{\mathbf{k}_j \in \mathcal{K}(i)} \mathcal{G}(D_{ij})}{1 + \sum_{\mathbf{k}_j \in \mathcal{K}(i)} \mathcal{G}(D_{ij}) + \sum_{\mathbf{k}_j \in \mathcal{N}(i)} \mathcal{G}(D_{ij})}, \tag{1}$$

where $D_{ij} = \mathbf{f}_j \cdot \mathbf{f}_i^{v_p} - \mathbf{f}_i^{v_q} \cdot \mathbf{f}_i^{v_p}$, $\mathcal{G}(x) = (1 + e^{-x/\tau})^{-1}$ is sigmoid function. To suppress error accumulation in sequential view matching, we propose a cyclic geometric consistency loss:

$$\mathcal{L}_{cgc} = 1 - \frac{1}{N} \sum_{p=1}^{N} \text{SmoothAP}(v_p \to v_{p \oplus 1}), \tag{2}$$

where

$$v_{p \oplus 1} = \begin{cases} v_{p+1}, & 1 \leq p < N, \\ v_1, & p = N. \end{cases}$$

Optimizing $\mathcal{L}_{cgc}$ minimizes ranking loss over a closed loop $v_1 \to v_2 \to \cdots \to v_N \to v_1$, reducing cumulative action estimation errors by aligning features from identical 3D location (Fig. 3 (b)). This mitigates view-specific biases and promotes viewpoint-invariant representation learning.

### 3.3 DYNAMIC-VIEW STREAM

Unlike static-view stream that relies on enduring and holistic scene comprehension, the dynamic-view stream prioritizes adaptive egocentric action reasoning. This visuomotor pipeline requires immediate exploitation of transient visual cues, emphasizing action-centric perception from egocentric viewpoints (Milner & Goodale, 2008).

Diverging from existing egocentric action prediction work (Dai et al., 2024; Plizzari et al., 2024) that focus on *what* actions occur, we infer *how* actions are executed by predicting kinematic parameters, including gripper translation, rotation, state (open or close) and collision indicator. Among these, gripper translation specifies 3D coordinates of end-effector, forming the geometric foundation for precise action proposals. In light of this, our dynamic-view stream achieves adaptive action reasoning by directly predicting end-effector position from a dynamic wrist-mounted camera view (*i.e.*, robot egocentric view). Accordingly, action reasoning is modeled as attention map generation, analogous to egocentric gaze estimation (Lai et al., 2024) that predicts human visual attention maps from first-person videos. This shared formulation enables seamless extraction of view-specific feature maps and saliency maps from gaze models. Both maps can be integrated into RVT-2, serving as dynamic-view features and heatmaps, respectively.

Fig. 2 illustrates the pipeline of dynamic-view stream, where a state-of-the-art egocentric gaze estimation model (GLC) (Lai et al., 2024) is utilized as feature extractor, coupled with RVT-2 action

head for action reasoning. First, we construct an egocentric video dataset through dynamic cameras, annotating each frame with ground-truth end-effector positions. Subsequently, we perform position-aware pretraining on this dataset, enabling GLC to infer positions from dynamic-view frames. During training of Cortical Policy, the pretrained GLC model remains frozen, while its intermediate representations (including feature maps and saliency maps) are extracted and fused with static-view counterparts for action decoding.

**Egocentric Video Rendering.** Preparing pretraining data requires addressing three critical issues: (1) Domain gap minimization: since human gaze provides localization cues for camera wearer actions (Li et al., 2018; Huang et al., 2020), the field-of-view (FOV) discrepancy between human head-mounted cameras (Grauman et al., 2022; Schaumlöffel et al., 2025) and robotic wrist-mounted cameras (James et al., 2020; Khazatsky et al., 2024) should be bridged, so as to transfer spatiotemporal localization priors from human gaze to end-effector position. (2) Positional invariance mitigation: the original wrist camera view produces invariant end-effector projections due to its fixed spatial relationship with end-effector, yielding non-informative annotations that increase overfitting risks and impede position-aware feature learning. (3) Cross-view alignment: the pretraining egocentric data serves as dynamic-view observations, thus needs to enable feature distribution consistency with static views to facilitate cross-view generalization. We resolve these issues by constructing dynamic virtual cameras within RVT renderer using real-time wrist camera extrinsics. Compared to raw wrist cameras, these virtual cameras adapt FOVs to match human egocentric data, diversify end-effector projections, and align processing with static viewpoints while preserving egocentric motion dynamics. The renderer associates each frame with its end-effector position via projection, generating annotated RGB-D sequences to constitute the final egocentric videos (see *supplementary material* for examples). In total, the dataset comprises 3,600 position-labeled videos (18 tasks $\times$ 100 episodes $\times$ 2 stages) at $224 \times 224$ resolution, exclusively used for position-aware pretraining.

**Position-aware Pretraining.** To enable knowledge transfer from human gaze estimation to end-effector position prediction, we initialize the GLC backbone with Ego4D-pretrained weights (Grauman et al., 2022), then fine-tune it on our egocentric video dataset. Each video is segmented into 5-second clips and resized to $256 \times 256$ resolution. Following Lai et al. (2024), we randomly sample 8 frames per clip to form input sequences. Each sequence is fed into GLC to generate spatiotemporally coherent saliency maps. For each frame, the end-effector location is determined by the most salient pixel in its saliency map. GLC ensures robust position localization through two core mechanisms: (i) capturing the temporal attention transition by leveraging egocentric motion cues in dynamic-view frames; (ii) explicitly modeling the spatial correlations between global and local tokens via its dedicated Global-Local Correlation module. Trained with KL-divergence loss for 15 epochs, we select the final GLC checkpoint for feature extraction in the dynamic-view stream.

**Dynamic-view Feature Extraction.** We extract intermediate representations from the pretrained GLC as action priors for training dynamic-view stream pipeline. For clarity, the GLC architecture is partitioned into Gaze Encoder (comprising Visual Token Embedding, Transformer Encoder and Global–Local Correlation modules) and Transformer Decoder. The Gaze Encoder outputs visual tokens that are projected as view feature maps via a trainable linear projection layer. The Transformer Decoder generates saliency maps as view heatmaps. Formally, given patch size $P$, batch size $B$, and GLC embedding dimension $D$, the dynamic-view feature map $\mathbf{F}$ is acquired by concatenating $\mathbf{F}^{SA} \in \mathbb{R}^{B \times (P \times P) \times D}$ from the last Transformer Encoder block and $\mathbf{F}^{GLC} \in \mathbb{R}^{B \times (P \times P) \times D}$ from the Global-Local Correlation module:

$$\mathbf{F} = \mathbf{LP}([\mathbf{F}^{SA}, \mathbf{F}^{GLC}]_c) \in \mathbb{R}^{B \times (P \times P) \times C}, \tag{3}$$

where the operator $[\cdot, \cdot]_c$ implements concatenation along the channel dimension, $\mathbf{LP}(\cdot)$ denotes linear projection that aligns the $2D$-dim GLC embeddings with RVT-2's $C$-dim token space, enabling integration of $\mathbf{F}$ into action decoding. With $D = 768$ and $P = 16$, our dual-stream transformer produces dynamic-view feature maps ($B \times 256 \times 1536$) and saliency maps ($B \times 1 \times 2 \times 128 \times 128$) via $2 \times 2 \times 2$ downsampling. For compatibility with static-view heatmaps, the saliency map is resized to $B \times 1 \times 2 \times 224 \times 224$, then temporally compressed to $B \times 1 \times 1 \times 224 \times 224$ via 3D convolution.

**Dual-stream Action Prediction.** Cortical Policy merges dual-stream outputs to determine gripper actions, where 3-DoF translation selects the highest-scoring 3D point from back-projected view heatmaps. For predicting 3-DoF rotation, gripper state and collision indicator, we follow Goyal et al. (2024) in leveraging both global and local features. In our implementation, four viewpoints are incorporated to represent the scene at time $t$, including three static views and one dynamic view.

Table 1: **Comparison with SOTA methods on RLBench.** The "Avg. Success" and "Avg. Rank" columns report the average success rate (%) and the average rank across 18 tasks. Best results are highlighted in bold, and the second best are underlined.

| Models | Reference | Avg. Success ↑ | Avg. Rank ↓ | Close Jar | Drag Stick | Insert Peg | Meat off Grill | Open Drawer | Place Cups | Place Wine | Push Buttons |
|---|---|---|---|---|---|---|---|---|---|---|---|
| Hiveformer | CoRL (2022) | 45.3 | 8.1 | 52 | 76 | 0 | **100** | 52 | 0 | 80 | 84 |
| PerAct | CoRL (2022) | 49.4 | 7.6 | 55.2±4.7 | 89.6±4.1 | 5.6±4.1 | 70.4±2.0 | 88.0±5.7 | 2.4±3.2 | 44.8±7.8 | 92.8±3.0 |
| RVT | CoRL (2023) | 62.9 | 5.7 | 52.0±2.5 | 99.2±1.6 | 11.2±3.0 | 88.0±2.5 | 71.2±6.9 | 4.0±2.5 | 91.0±5.2 | 100.0±0.0 |
| Σ-agent | CoRL (2024) | 68.8 | 4.2 | 78.4±2.9 | 100.0±0.0 | 15.2±2.9 | 97.6±1.9 | 76.8±3.8 | 0.8±1.3 | 90.4±3.5 | 100.0±0.0 |
| SAM-E | ICML (2024) | 70.6 | 3.8 | 82.4±3.6 | 100.0±0.0 | 18.4±4.6 | 95.2±3.3 | 95.2±5.2 | 0.0±0.0 | 94.4±4.6 | 100.0±0.0 |
| VIHE | IROS (2024) | 77 | 3.6 | 48 | **100** | **84** | **100** | 76 | 12 | 88 | **100** |
| RVT-2 | RSS (2024) | 77.5 | 3.5 | 93.3±1.9 | 97.3±1.9 | 28.0±3.3 | 100.0±0.0 | 92.0±3.3 | 32.0±5.7 | 84.0±9.8 | 100.0±0.0 |
| 3D-MVP | CVPR (2025) | 67.5 | 4.3 | 76.0 | 100.0 | 20.0 | 96.0 | 84.0 | 4.0 | 100.0 | 96.0 |
| Ours | – | **81.0** | **1.8** | **96.0**±0.0 | 100.0±0.0 | 38.7±6.8 | 100.0±0.0 | 84.0±6.5 | 24.0±3.3 | 94.7±3.8 | 100.0±0.0 |

| Models | Reference | Put in Cupboard | Put in Drawer | Put in Safe | Screw Bulb | Slide Block | Sort Shape | Stack Blocks | Stack Cups | Sweep to Dustpan | Turn Tap |
|---|---|---|---|---|---|---|---|---|---|---|---|
| Hiveformer | CoRL (2022) | 32 | 68 | 76 | 8 | 64 | 8 | 8 | 0 | 28 | 80 |
| PerAct | CoRL (2022) | 28.0±4.4 | 51.2±4.7 | 84.0±3.6 | 17.6±2.0 | 74.0±13.0 | 16.8±4.7 | 26.4±3.2 | 2.4±2.0 | 52.0±0.0 | 88.0±4.4 |
| RVT | CoRL (2023) | 49.6±3.2 | 88.0±5.7 | 91.2±3.0 | 48.0±5.7 | 81.6±5.4 | 36.0±2.5 | 28.8±3.9 | 26.4±8.2 | 72.0±0.0 | 93.6±4.1 |
| Σ-agent | CoRL (2024) | 66.4±4.5 | 70.4±3.8 | 98.4±1.9 | 73.2±2.2 | 74.4±4.5 | 36.0±3.2 | 51.2±5.4 | 33.6±6.7 | 80.8±1.3 | 95.2±1.3 |
| SAM-E | ICML (2024) | 64.0±2.8 | 92.0±5.7 | 95.2±3.3 | 78.4±3.6 | 95.2±1.8 | 34.4±6.1 | 26.4±4.6 | 0.0±0.0 | 100.0±0.0 | 100.0±0.0 |
| VIHE | IROS (2024) | 60 | 96 | 92 | **92** | **96** | **52** | 68 | 68 | 64 | 92 |
| RVT-2 | RSS (2024) | 44.0±6.5 | 98.7±1.9 | 92.0±3.3 | 86.7±1.9 | 74.7±5.0 | 26.7±1.9 | 80.0±5.7 | 72.0±0.0 | 98.7±1.9 | 94.7±1.9 |
| 3D-MVP | CVPR (2025) | 60.0 | **100.0** | 92.0 | 60.0 | 48.0 | 28.0 | 40.0 | 36.0 | 80.0 | 96.0 |
| Ours | – | 65.3±9.4 | 100.0±0.0 | 98.7±1.9 | 81.3±1.9 | 86.7±1.9 | 37.3±1.9 | 81.3±1.9 | 76.0±3.3 | 100.0±0.0 | 94.7±5.0 |

Each viewpoint predicts a feature map $\mathbf{F}_j$ and a heatmap $\mathbf{H}_j$ that indicates the end-effector pixel coordinate. Local features are pooled from $\mathbf{F}_j$ at these coordinates, while the global feature vector is formed by concatenating the following components:

$$[ \underbrace{\phi(\mathbf{F}_1 \odot \mathbf{H}_1); \phi(\mathbf{F}_2 \odot \mathbf{H}_2); \phi(\mathbf{F}_3 \odot \mathbf{H}_3);}_{\text{static views}} \underbrace{\phi(\mathbf{F}_4 \odot \mathbf{H}_4);}_{\text{dynamic view}} \underbrace{\psi(\mathbf{F}_1); \psi(\mathbf{F}_2); \psi(\mathbf{F}_3);}_{\text{static views}} \underbrace{\psi(\mathbf{F}_4)}_{\text{dynamic view}} ],$$

where $\odot$ is element-wise multiplication; $\phi(\cdot)$ and $\psi(\cdot)$ represent sum and max-pooling, respectively. The GLC-generated representations ensure that $\mathbf{H}_4$ effectively highlights task-relevant egocentric cues (*e.g.*, end-effector positions), thereby producing highly focused global features through heatmap-weighting rule $\mathbf{F}_j \odot \mathbf{H}_j$. The total loss combines action prediction loss and cross-view geometric consistency loss in Eq. (2):

$$\mathcal{L} = \mathcal{L}_{action} + \lambda\mathcal{L}_{cgc}, \tag{4}$$

where $\mathcal{L}_{action}$ is defined as the sum of cross-entropy losses for each action component, and $\lambda$ is a trade-off parameter set to 1. Through optimizing Eq. (4), Cortical Policy unifies dynamic-view action cues and static-view spatial knowledge, enabling the policy to robustly adapt to environmental perturbations and dynamic scene changes.

## 4 EXPERIMENT

This section evaluates Cortical Policy by answering the following questions: (1) How well does Cortical Policy perform in manipulation compared to state-of-the-art policies? (2) What impact do geometric consistency loss and various design choices in dynamic-view stream have on overall performance? (3) How robust is Cortical Policy against environmental perturbations (*e.g.*, distractors, changes in camera pose, and object properties)? (4) Does Cortical Policy work in real-world tasks? To this end, we conduct experiments in both simulation and real-world scenarios, reporting results in Sections 4.2, 4.3, 4.4, 4.5 respectively.

### 4.1 EXPERIMENTAL SETUP

We begin with an overview of the datasets, baselines and evaluation metrics. For more detailed experimental settings, please refer to Appendix B.

**Benchmark Datasets.** For fair comparison, we adopt a standard multi-task manipulation benchmark that contains 18 RLBench (James et al., 2020) tasks with 249 language-specified variations

Table 2: **Ablation study on dual-stream view transformer.** All designs contribute to improving performance of Cortical Policy. "Arch.", "Pre.", "Heat." denote model architecture, position-aware pretraining, dynamic-view heatmap, respectively. "Single" means single-stream model with only static viewpoints, "Dual" means dual-stream model integrating dynamic and static viewpoints.

| Models | Arch. | $\mathcal{L}_{cgc}$ | Pre. | Heat. | Avg. Success↑ | Avg. Rank↓ | Close Jar | Drag Stick | Insert Peg | Meat off Grill | Open Drawer | Place Cups | Place Wine | Push Buttons |
|---|---|---|---|---|---|---|---|---|---|---|---|---|---|---|
| **A** | Single | ✗ | – | – | 77.5 | 3.3 | 93.3±1.9 | 97.3±1.9 | 28.0±3.3 | **100.0**±0.0 | 92.0±3.3 | 32.0±5.7 | 84.0±9.8 | **100.0**±0.0 |
| **B** | Single | ✔ | – | – | 80.1 | 2.4 | 94.7±1.9 | **100.0**±0.0 | 25.3±5.0 | **100.0**±0.0 | 94.7±1.9 | 21.3±1.9 | 86.7±5.0 | **100.0**±0.0 |
| **C** | Dual | ✗ | ✗ | ✔ | 77.6 | 3.0 | **97.3**±0.0 | 98.7±0.0 | 30.7±10.0 | **100.0**±0.0 | 88.0±3.3 | 28.0±6.5 | 93.3±5.0 | **100.0**±0.0 |
| **D** | Dual | ✗ | ✔ | ✗ | 73.3 | 4.8 | 90.7±1.9 | 90.7±13.2 | 26.7±3.8 | **100.0**±0.0 | 94.7±1.9 | 20.0±0.0 | 82.7±8.2 | 98.7±1.9 |
| **E** | Dual | ✗ | ✔ | ✔ | 79.5 | 3.1 | 90.7±1.9 | 97.3±1.9 | 29.3±1.9 | **100.0**±0.0 | **100.0**±0.0 | **46.7**±7.5 | 88.0±5.7 | **100.0**±0.0 |
| **F (Ours)** | Dual | ✔ | ✔ | ✔ | **81.0** | **1.9** | 96.0±0.0 | **100.0**±0.0 | **38.7**±6.8 | **100.0**±0.0 | 84.0±6.5 | 24.0±3.3 | **94.7**±3.8 | **100.0**±0.0 |

| Models | Arch. | $\mathcal{L}_{cgc}$ | Pre. | Heat. | Put in Cupboard | Put in Drawer | Put in Safe | Screw Bulb | Slide Block | Sort Shape | Stack Blocks | Stack Cups | Sweep to Dustpan | Turn Tap |
|---|---|---|---|---|---|---|---|---|---|---|---|---|---|---|
| **A** | Single | ✗ | – | – | 44.0±6.5 | 98.7±1.9 | 92.0±3.3 | 86.7±1.9 | 74.7±5.0 | 26.7±1.9 | 80.0±5.7 | 72.0±0.0 | 98.7±1.9 | 94.7±1.9 |
| **B** | Single | ✔ | – | – | 61.3±5.0 | 98.7±1.9 | 97.3±1.9 | **92.0**±5.7 | 82.7±1.9 | 18.7±1.9 | **92.0**±3.3 | **81.3**±5.0 | **100.0**±0.0 | 94.7±1.9 |
| **C** | Dual | ✗ | ✗ | ✔ | **73.3**±5.0 | 96.0±0.0 | 92.0±0.0 | 85.3±6.8 | 78.7±5.0 | 6.7±3.8 | 81.3±1.9 | 50.7±5.0 | **100.0**±0.0 | **96.0**±0.0 |
| **D** | Dual | ✗ | ✔ | ✗ | 48.0±14.2 | 97.3±3.8 | 88.0±8.6 | 85.3±1.9 | 65.3±5.0 | 18.7±10.5 | 86.7±1.9 | 44.0±14.2 | 94.7±5.0 | 88.0±11.8 |
| **E** | Dual | ✗ | ✔ | ✔ | 50.7±6.8 | 88.0±3.3 | 89.3±1.9 | 88.0±6.5 | 84.0±3.3 | 22.7±3.8 | 82.7±3.8 | **81.3**±6.8 | 98.7±1.9 | 93.3±6.8 |
| **F (Ours)** | Dual | ✔ | ✔ | ✔ | 65.3±9.4 | **100.0**±0.0 | **98.7**±1.9 | 81.3±1.9 | **86.7**±1.9 | **37.3**±1.9 | 81.3±1.9 | 76.0±3.3 | **100.0**±0.0 | 94.7±5.0 |

simulated in CoppeliaSim (Rohmer et al., 2013). The tasks are performed by a Franka Panda robot arm with a parallel jaw gripper. Raw visual observations are captured by four $128 \times 128$ RGB-D cameras mounted at front, left shoulder, right shoulder, and wrist of the robot. The policy-predicted gripper poses are executed by a sampling-based motion planner. Each behavior-cloning agent is allowed up to 25 steps to complete a task. Following PerAct, we use the same training-test split, training all models on 100 demonstrations per task and evaluating a single checkpoint on all tasks.

**Baselines and Evaluation Metrics.** We benchmark Cortical Policy against 8 state-of-the-art manipulation policies: Hiveformer (Guhur et al., 2022), PerAct (Shridhar et al., 2023), RVT (Goyal et al., 2023), VIHE (Wang et al., 2024), RVT-2 (Goyal et al., 2024), $\Sigma$-agent (Ma et al., 2024), SAM-E (Zhang et al., 2024), and 3D-MVP (Qian et al., 2025), which are predominantly based on view transformer architectures and have demonstrated effectiveness in 3D object manipulation. For visual input, PerAct uses 3D voxels, Hiveformer utilizes raw cameras positioned on the wrist and both shoulders, whereas the other baselines employ multiple static virtual cameras. We report success rates for individual tasks, along with average success rate and rank across all tasks.

## 4.2 Performance comparison on RLBench

Table 1 summarizes the comparison results on RLBench. Cortical Policy achieves the highest average success rate, outperforming the best-performing baseline (RVT-2) by an absolute improvement of 3.5%. In terms of individual tasks, our model achieves top-1 or top-2 performance in 14 out of 18 tasks. These results demonstrate the efficacy of Cortical Policy for robotic manipulation, advancing toward human-like visuomotor control. For tasks where RVT and RVT-2 already achieve success rates above 90%, our dual-stream framework generally yields better performance, as seen in "close jar", "sweep to dustpan", and "put in safe". We observe that our model outperforms existing methods in multi-object tasks, such as "stack cups" and "stack blocks", with a margin of 1.3%-4.0%. These tasks implicitly require understanding spatial relationships among objects, validating the effectiveness of 3D prior injection in the static-view stream.

## 4.3 Ablation study

To evaluate the impact of key design choices in Cortical Policy, we conduct ablation experiments on RLBench, with results summarized in Table 2. The ablated variants are implemented as: **(A)** Removing the entire dynamic-view stream along with cross-view geometric consistency loss $\mathcal{L}_{cgc}$. **(B)** Using only static-view stream. **(C)** Ablating position-aware pretraining and instead fine-tuning the gaze model jointly with manipulation policy in an end-to-end manner, excluding $\mathcal{L}_{cgc}$. **(D)** Employing only view feature maps without heatmaps during dynamic-view feature extraction, also excluding $\mathcal{L}_{cgc}$. **(E)** Removing $\mathcal{L}_{cgc}$ while retaining all components of dynamic-view stream. An identical training configuration is maintained for all ablation studies. The discussion follows.

**Effects of cross-view geometric consistency.** $\mathcal{L}_{cgc}$ leads to consistent improvements across architectures, *e.g.*, variant **B** outperforms **A** by 2.6%, the full model **F** surpasses **E** by 1.5%, demonstrat-

Table 3: **Results on THE COLOSSEUM.** The "Avg. Success" and "Avg. Rank" columns report the average success rate (%) and the average rank across all perturbations on 4 COLOSSEUM tasks.

| Models | Arch. | $\mathcal{L}_{cgc}$ | Pre. | Heat. | Avg. Success↑ | Avg. Rank↓ | All Perturbations | MO-Color | RO-Color | MO-Texture | RO-Texture | MO-Size |
|---|---|---|---|---|---|---|---|---|---|---|---|---|
| PerAct | – | – | – | – | 7.7 | 7.0 | 0.0±0.0 | 8.0±8.5 | 5.3±5.0 | 2.0±2.0 | 4.0±3.3 | 16.0±17.3 |
| RVT | – | – | – | – | 37.7 | 6.0 | 3.0±5.2 | 27.0±27.3 | 36.0±15.0 | 50.0±38.0 | 57.3±32.7 | 50.7±29.6 |
| RVT-2 | Single | ✗ | – | – | 60.5 | 4.4 | **15.0**±17.3 | 64.0±25.6 | 64.9±27.2 | 93.4±2.7 | 66.2±31.8 | 80.4±17.5 |
| Variant **B** | Single | ✔ | – | – | 63.8 | 3.3 | 10.3±8.0 | 69.7±28.1 | 70.2±28.0 | 95.4±0.7 | 71.6±30.9 | 84.0±14.2 |
| Variant **D** | Dual | ✗ | ✔ | ✗ | 66.4 | 2.9 | 10.0±8.2 | 69.7±27.5 | 72.9±29.9 | 94.0±2.0 | **74.7**±24.5 | 83.6±16.7 |
| Variant **E** | Dual | ✗ | ✔ | ✔ | 68.7 | 2.4 | 8.7±15.0 | 75.0±29.7 | 73.8±29.6 | 96.7±0.7 | 71.1±33.4 | 82.7±13.6 |
| Ours | Dual | ✔ | ✔ | ✔ | **69.9** | **1.9** | 10.0±15.1 | **78.0**±26.9 | **76.9**±28.9 | **100.0**±0.0 | 66.7±27.8 | **86.7**±16.1 |

| Models | Arch. | $\mathcal{L}_{cgc}$ | Pre. | Heat. | RO-Size | Light Color | Table Color | Table Texture | Distractor | Background Texture | RLBench Variations | Camera Pose |
|---|---|---|---|---|---|---|---|---|---|---|---|---|
| PerAct | – | – | – | – | 9.3±1.9 | 7.0±4.4 | 8.0±6.3 | 3.0±3.3 | 2.7±3.8 | 8.0±6.9 | 25.0±24.7 | 9.0±7.1 |
| RVT | – | – | – | – | 22.7±29.3 | 52.0±30.7 | 42.0±30.8 | 48.0±27.9 | 13.3±13.2 | 40.0±32.9 | 41.0±26.6 | 45.0±31.4 |
| RVT-2 | Single | ✗ | – | – | 44.4±28.2 | 63.7±30.4 | 42.3±31.7 | 54.4±24.4 | 60.4±30.5 | 72.0±27.9 | 63.7±27.0 | 62.7±28.6 |
| Variant **B** | Single | ✔ | – | – | 44.0±33.1 | 73.7±26.4 | 47.3±30.7 | 61.0±25.2 | 64.0±31.2 | 67.0±24.6 | 68.0±25.2 | 66.7±31.4 |
| Variant **D** | Dual | ✗ | ✔ | ✗ | 40.0±27.9 | 73.2±26.2 | 49.4±31.1 | 69.7±25.1 | 79.1±22.0 | 70.0±26.6 | 74.7±22.9 | 68.3±32.2 |
| Variant **E** | Dual | ✗ | ✔ | ✔ | **53.8**±27.2 | **78.3**±23.7 | **60.0**±25.1 | 70.7±25.9 | 74.2±18.5 | 68.7±24.0 | 76.7±22.3 | 71.4±30.9 |
| Ours | Dual | ✔ | ✔ | ✔ | 51.6±33.5 | 69.3±34.6 | 46.7±32.4 | **75.0**±27.5 | **83.1**±20.1 | **77.7**±26.9 | **82.3**±17.8 | **74.0**±32.7 |

ing the effectiveness of $\mathcal{L}_{cgc}$ for both single-stream and dual-stream policies. This validates that our viewpoint-invariant representation learning method benefits robotic manipulation.

**Effects of position-aware pretraining.** Compared to end-to-end training (variant **C**), freezing position-aware pretrained gaze model (variant **E**) obtains 1.9% higher average success rate and stability across tasks. This demonstrates the superiority of our pretraining approach.

**Choice of gaze model representations.** Our dynamic-view stream utilizes both feature maps and heatmaps from the gaze model. Without heatmaps, variant **D** underperforms single-stream variants, confirming that the heatmaps' explicit action cues are crucial to dynamic-view stream.

**Dual-stream versus single-stream architecture.** Both the static-view and dynamic-view streams boost performance, with gains of 2.6% (variant **B** vs. **A**) and 0.9% (variant **F** vs. **B**), respectively. This demonstrates the effectiveness of incorporating dynamic-view perception for action prediction. Notably, the dynamic virtual camera breaks the strict orthographic constraints of multi-camera setups in existing view transformers, while it still improves performance. We also record computational time for each component (Fig. 4 (a)), showing that our dual-stream design enhances performance without sacrificing efficiency.

## 4.4 ROBUSTNESS EVALUATION ON COLOSSEUM

We further evaluate the robustness and generalization capabilities of our method on the COLOS-SEUM benchmark (Pumacay et al., 2024), which is an extension of RLBench. The models trained on the original RLBench tasks are evaluated in environments spanning diverse unseen perturbations, encompassing changes in object color and size, lighting, distractors, and camera poses, *etc*. As shown in Table 3, Cortical Policy obtains the highest average success rate among all the baselines (including PerAct, RVT-2 and its ablation variants), notably outperforming RVT-2 by 9.4%. Crucially, ablation results indicate that the **dynamic-view stream** is the primary driver of this robustness, contributing significantly larger gains than $\mathcal{L}_{cgc}$ under severe perturbations. Among all the 14 evaluated generalization settings, our method achieves the top performance in 9 of them. These results demonstrate that Cortical Policy possesses strong robustness against environmental perturbations. More details about the data and results of the COLOSSEUM benchmark are in Appendix G.

## 4.5 REAL-WORLD EXPERIMENT

We design four real-world tasks for evaluation: a basic task ("stack 2 blocks") aligned with the RLBench "stack blocks" task, a spatial reasoning task ("stack 2 blocks in between the bottles") and two challenging dynamic tasks ("stack 2 blocks with target/base displacement"). These tasks extend the simulated stacking scenario by introducing real-world complexities including spatial constraints and unpredictable scene dynamics. Each task is evaluated through 10 trials (see Appendix B.2 for hardware details). As shown in Fig. 4 (b), compared to ablated variants (**B**, **E**), RVT, RVT-2, and 3D-

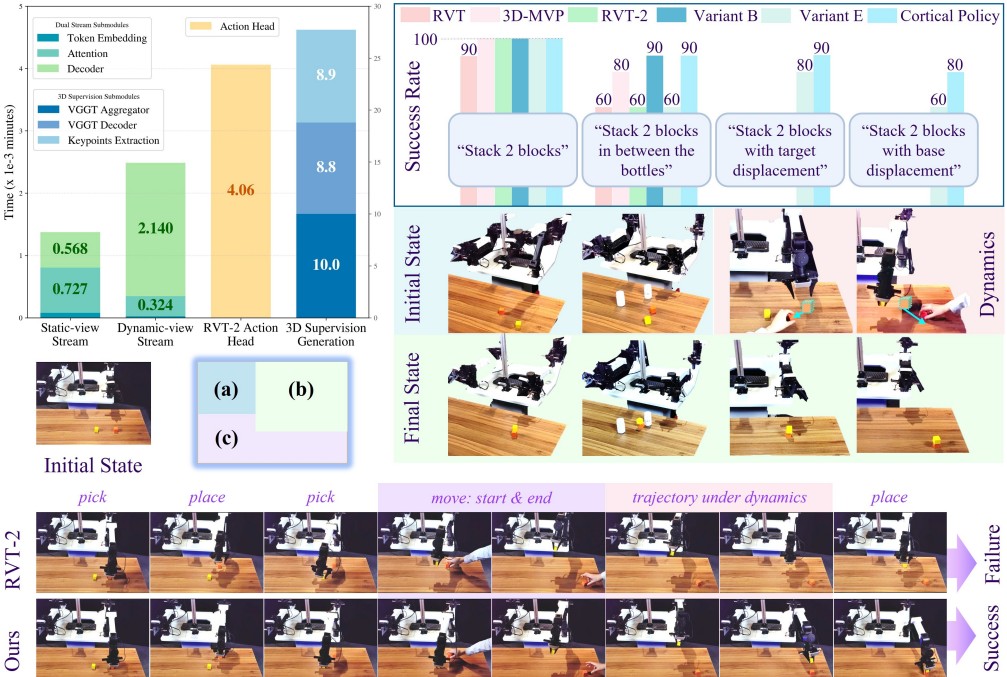

Figure 4: (a) **Training time of Cortical Policy modules**, with time cost of 3D supervision generation, dual streams, action head. (b) (Top) **Real-world performance comparison.** (Bottom) Visualization of the initial and final states for the four real-world tasks. (c) **Trajectory visualization for "stack 2 blocks with base displacement" task.**

MVP, Cortical Policy achieves: (1) a 30% higher success rate than RVT and RVT-2 (and 10% over 3D-MVP) in the spatial reasoning task, confirming that $\mathcal{L}_{cgc}$ enhances geometric understanding; (2) an 80% success rate under dynamic perturbations, whereas static-view-only approaches completely fail (0%). Fig. 4(c) demonstrates that our method succeeds by dynamically re-planning trajectories, while baselines fail to do so, highlighting Cortical Policy's adaptation capability through dynamic-view processing. These real-world results collectively validate the robustness and superiority of our method in physical deployment. Demos can be found in supplementary material.

## 5 CONCLUSION

This paper presents Cortical Policy, a dual-stream framework for enhancing spatial reasoning and dynamic-scene adaptability of robotic manipulation policies. Through VGGT-supervised geometric consistency optimization, we inject strong 3D priors into the policy, thereby improving spatial understanding. Complementing this, the dynamic-view stream *learns to discover and attend to action-critical targets*, demonstrating its effectiveness in tracking the end-effector. This enables adaptive adjustment to task dynamics, an ability absent in prior work. Extensive experiments demonstrate the superiority of Cortical Policy in both simulated and real-world scenarios, highlighting the contribution of the dynamic-view stream to handling unpredictable scene perturbations.

**Limitations and Future Work.** While Cortical Policy demonstrates strong within-task generalization (validated on COLOSSEUM), its zero-shot transfer to novel tasks remains challenging, as reflected by the 24% success rate on the unseen "close laptop lid" task. A promising direction is to enhance its compositional abstraction capability for task generalization (*e.g.*, by recombining learned perceptual and motor primitives). Building on its modular design, we plan to extend this framework with multi-resolution encoders and hierarchical attention mechanisms to handle extremely fine-grained manipulation. The adaptive fusion of dual-stream representations at token and viewpoint levels also warrants further exploration. Furthermore, extending dynamic-view stream to track diverse targets beyond the end-effector (*e.g.*, specific objects, affordance points, multiple entities) will further probe the framework's generalization in open-world settings.

REPRODUCIBILITY STATEMENT

To ensure reproducibility of Cortical Policy, our implementation details are provided in Appendix B.3. Sections 3.2 and 3.3 describe the methodology and data processing steps for egocentric video dataset used in position-aware pretraining. Additionally, the anonymous source code is available in supplementary material to facilitate validation and replication of our findings.

ACKNOWLEDGMENTS

We would like to thank the reviewers for their constructive comments. This work is supported by National Natural Science Foundation of China (Grant No. 62406092), National Natural Science Foundation of China (Grant No. U24B20175), Shenzhen Science and Technology Program (Grant No. KJZD20240903100017022), Guangdong Basic and Applied Basic Research Foundation (Grant No. 2025A1515010169), Shenzhen Science and Technology Program (Grant No. KQTD20240729102207002).

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

APPENDIX

This appendix provides supplementary materials supporting the main paper, organized as follows:

- **LLM Usage Disclosure**: Role specification of large language models
- **Experimental Setup**: RLBench tasks, implementation details and baselines
- **Time Analysis and Capacity Analysis**: Additional computational efficiency analysis
- **Keypoints Visualization**: Qualitative results of geometrically consistent keypoints
- **Egocentric Rendering Visualization**: Position-aware pretraining data samples
- **Failure Case Analysis**: In-depth investigation of RVT-2's spatial reasoning limitations
- **COLOSSEUM Experiments**: Comprehensive generalization and robustness evaluation

## A  LARGE LANGUAGE MODEL USAGE DISCLOSURE

In compliance with ICLR 2026 policy, we disclose the use of large language models (LLMs) in the preparation of this work:

- DeepSeek-R1 (`https://www.deepseek.com`) was utilized exclusively for **language polishing** of non-technical sections (Introduction and Related Work).
- All technical content (Method, Experiment and Conclusion) was written by humans without LLM assistance.
- LLM-generated text was rigorously verified and modified by the authors.
- No LLM was used for data analysis, algorithm design, or scientific interpretation.

The authors assume full responsibility for all content in this manuscript.

## B  EXPERIMENTAL SETUP

This section specifies the experimental framework covering RLBench tasks, real-robot setup, our implementation details, baseline architectures and processing pipelines.

### B.1  RLBENCH TASKS

We briefly summarize the RLBench tasks in Table 4, comprising 18 tasks with 249 variations across object color, category, placement, count, shape, and size. Each task requires executing manipulation sequences such as pick-and-place, tool use, drawer opening, and precision operations like peg insertion and shape sorting. During evaluation, the robot handles variations including novel object poses, randomly sampled language instructions, and scenes with unseen object appearances. This task variety necessitates manipulation policies with generalizable comprehension of scenes and instructions, along with adaptable skill acquisition beyond specialized adaptation to individual scenarios.

For a more detailed introduction of each task, please refer to PerAct (Shridhar et al., 2023). For training and evaluating Cortical Policy, we render four virtual camera views to get visual inputs, including 3 static viewpoints and 1 dynamic viewpoint. We visualize the rendered images in Fig. 5.

### B.2  REAL-ROBOT EXPERIMENTAL SETUP

To evaluate Cortical Policy in real-world scenarios, we deploy a tabletop manipulation system consisting of a dual-arm Cobot Agilex ALOHA robot. As shown in Fig. 6, the experimental setup integrates two fixed cameras for static-view perception, complemented by two wrist-mounted cameras for dynamic-view perception. In our experiments, we utilize a single arm to execute four distinct manipulation tasks: one benchmark task aligned with RLBench and three new tasks designed to test spatial reasoning and dynamic scene adaptation abilities. Each task collects 45 human-teleoperated demonstrations with placement variations, and a single agent is trained in a multi-task setting on all four tasks. For evaluation, this unified agent is tested on novel spatial configurations unseen in the training demonstrations. Four real-world tasks are detailed as follows:

Table 4: Summary of the 18 RLBench tasks for multi-task experiments.

| Task Name | Language Template | #of Variations | Variation Type |
|---|---|---|---|
| close jar | "close the __ jar" | 20 | color |
| drag stick | "use the stick to drag the cube onto the __ target" | 20 | color |
| insert peg | "put the ring on the __ spoke" | 20 | color |
| meat off grill | "take the __ off the grill" | 2 | category |
| open drawer | "open the __ drawer" | 3 | placement |
| place cups | "place __ cups on the cup holder" | 3 | count |
| place wine | "stack the wine bottle to the __ of the rack" | 3 | placement |
| push buttons | "push the __ button, [then the __ button]" | 50 | color |
| put in cupboard | "put the __ in the cupboard" | 9 | category |
| put in drawer | "put the item in the __ drawer" | 3 | placement |
| put in safe | "put the money away in the safe on the __ shelf" | 3 | placement |
| screw bulb | "screw in the __ light bulb" | 20 | color |
| slide block | "slide the block to __ target" | 4 | color |
| sort shape | "put the __ in the shape sorter" | 5 | shape |
| stack blocks | "stack __ __ blocks" | 60 | color, count |
| stack cups | "stack the other cups on top of __ the cup" | 20 | color |
| sweep to dustpan | "sweep dirt to the __ dustpan" | 2 | size |
| turn tap | "turn __ tap" | 2 | placement |

- **Stack 2 blocks**: This basic task requires the robot to sequentially stack a yellow block onto an orange block, corresponding to RLBench "stack blocks" task for sim-to-real transfer evaluation.
- **Stack 2 blocks in between the bottles**: This task is an extended version of the basic task, testing the understanding of spatial relationships by requiring the robot to: (1) Precisely place the orange block in the region between two bottles. (2) Stably stack the yellow block atop the orange block.
- **Stack 2 blocks with target displacement**: This task introduces real-world unpredictability, evaluating how effectively the dynamic-view stream handles trajectory adaptation. While the robot is approaching the first block, it is displaced, requiring adaptive trajectory re-planning to complete the original stacking task.
- **Stack 2 blocks with base displacement**: This task also tests the dynamic adaptation capability by displacing the already-stacked orange block during the yellow block's approach phase. The robot must re-locate the orange block and put the yellow block on it.

For each real-world task, 10 independent trials are conducted to calculate the overall success rate. A trial was considered successful only if all sub-actions of the task are executed correctly.

### B.3 IMPLEMENTATION DETAILS

All models are trained on 8 NVIDIA A800 GPUs. We measure the training efficiency of Cortical Policy on an NVIDIA A800 GPU, revealing that VGGT-based 3D supervision generation constitutes the most computationally intensive component. As shown in Fig. 4 (a), the average time costs for VGGT feature aggregation, VGGT decoding, and geometrically consistent keypoint extraction are $1.00 \times 10^{-2}$, $8.80 \times 10^{-3}$, and $8.92 \times 10^{-3}$ minutes respectively, resulting in a total of $3.09 \times 10^{-2}$ minutes for the complete 3D supervision generation. This process is $4.7\times$ slower than the action reasoning procedure of Cortical Policy, primarily due to the computational demands of VGGT inference. To mitigate this bottleneck, we implement a multi-stage strategy that decouples 3D supervision generation from feature consistency optimization. Specifically, geometrically consistent keypoints are precomputed from VGGT, stored, and indexed by their corresponding demonstration IDs.

Cortical Policy is trained for 32.5K steps using the 8-bit LAMB optimizer (Dettmers et al., 2022) with a cosine learning rate decay schedule and 2K-step warm-up. We select the final converged model for evaluation. For baseline methods excluding RVT-2, we report evaluation results from their original publications, with the performance of Hiveformer reported by Chen et al. (2023), and the performance of PerAct reported by Goyal et al. (2023). Given the architectural similarities between our framework and RVT-2, we conduct a controlled comparison by training RVT-2 from scratch under identical conditions as ours, including the same computing resources and matching

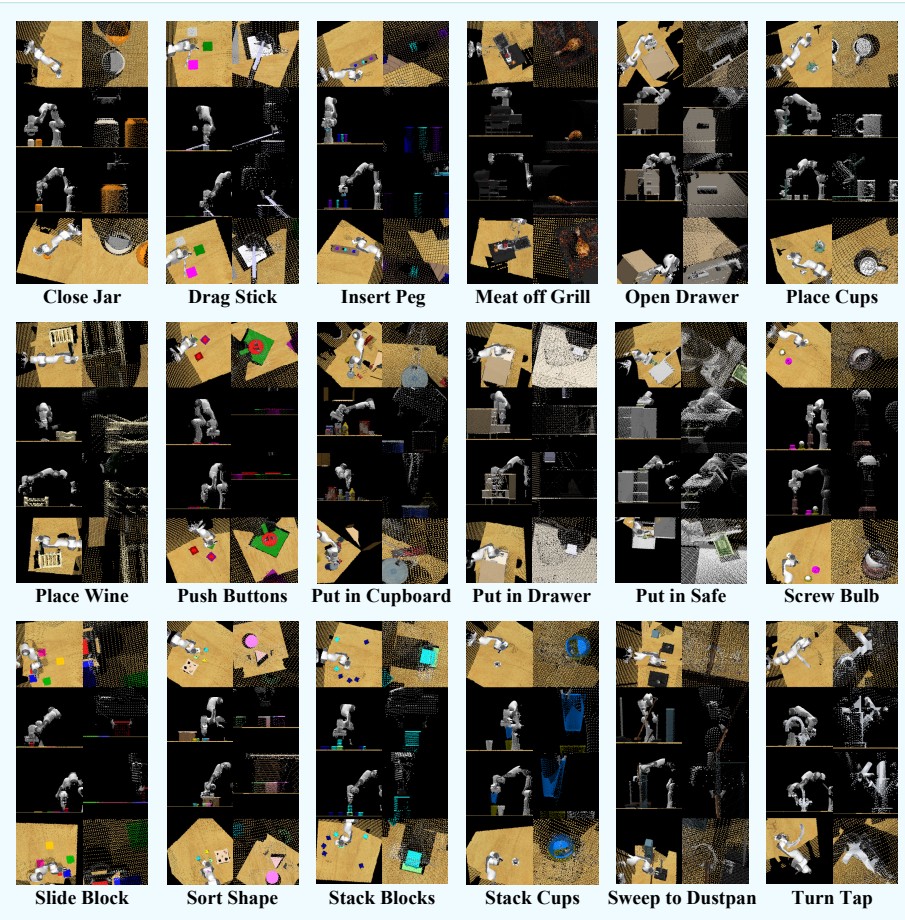

Figure 5: **Rendered views for 18 RLBench tasks.** Three orthographic static cameras and a dynamic camera (defined by the wrist camera's raw extrinsic parameters) are used to generate image inputs. For each task, the first three lines show the static views (top, front, right), and the last line shows the dynamic view; rendered results of the first (coarse) stage are presented in the left part while that of the second (fine) stage are shown right.

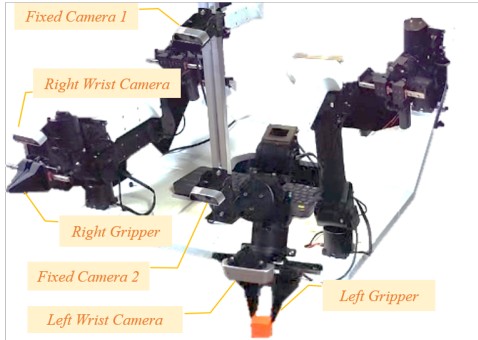

Figure 6: Real-world setup.

hyperparameters (32.5K training steps with batch size 512). This implementation differs from the pretrained RVT-2 model (∼80K training steps with batch size 192) released by Goyal et al. (2024), ensuring a fair assessment of our methodological contributions. To account for the randomness in RLBench's sampling-based motion planner, we perform three independent test runs per model, each

Table 5: Training Hyperparameters of Cortical Policy.

| Hyperparameters | Value |
|---|---|
| Batch size | 512 |
| Learning rate | $5.44 \times 10^{-3}$ |
| Optimizer | LAMB |
| Learning rate schedule | cosine decay |
| Weight decay | $1 \times 10^{-4}$ |
| Warm-up steps | 2000 |
| Training steps | 32.5K |
| Training epochs | 104 |
| $\mathcal{L}_{cgc}$ loss weight ($\lambda$) | 1 |
| Negative set distance threshold ($\zeta$) | 0.1 |
| Keypoints per view ($M$) | 300 |
| Sigmoid temperature ($\tau$) | 0.01 |
| Number of static views ($N$) | 3 |
| GLC training epochs | 15 |

comprising 25 episodes per task. The resulting average success rates with standard deviations are reported in Tables 1 and 2.

We implement data augmentation protocols consistent with established view transformers (Goyal et al., 2023; 2024). For translational augmentation, point clouds are randomly perturbed within $\pm 12.5\,\text{cm}$ along each Cartesian axis. For rotational augmentation, point clouds undergo random $z$-axis rotations bounded by $\pm 45°$. Table 5 details our training configuration, including a batch size of 512 ($64 \times 8$) and a learning rate scaling with batch size as $1.0625 \times 10^{-5} \times \text{bs}$.

## B.4   BASELINES

This section details the baseline manipulation policies that are based on view transformers, analyzing their view processing architectures and vision-to-action mapping frameworks.

(1) Hiveformer (Guhur et al., 2022) predicts actions conditioned on a natural language instruction, visual observations at $t$ steps (RGB images, point clouds and proprioception from wrist, left shoulder, and right shoulder cameras) and previous actions at $t$ steps (gripper translation, rotation, and open/close state). Multi-modal tokens are formed by concatenating word tokens and visual tokens from all camera views with embeddings of camera ID, step ID, modality type, and patch location. A transformer encoder then models relationships among camera views, observations and instructions, current and history information. Finally, a CNN decoder predicts rotation and gripper state, while a UNet decoder predicts translation.

(2) RVT (Goyal et al., 2023) re-renders original visual observations (RGB-D images from front, left shoulder, right shoulder, and wrist cameras) into five static virtual viewpoints anchored at the robot base (front, top, left, right, back). This generates 7-channel images: 3 for RGB, 1 for depth, and 3 for pixel coordinates. These re-rendered images, along with language instruction and gripper state, are processed by a joint transformer that sequentially computes intra-view attention, cross-view attention and vision-language attention. The model outputs view-specific heatmaps for predicting 3D translation, and outputs global features that concatenate all viewpoints for estimating gripper rotation, state, and collision indicator.

(3) VIHE (Wang et al., 2024) employs a multi-stage view rendering and action refinement framework comprising an initial global stage and two refinement stages. The initial stage replicates RVT's five-camera rendering, while the subsequent stages autoregressively generate five virtual in-hand views attached to the previously predicted gripper pose, enabling progressively finer workspace focus. The view transformer adopts masked self-attention to facilitate intra-stage and cross-stage interactions among language instructions, proprioception, multi-stage and multi-camera tokens. During refinement, relative transformations are predicted to update prior stage outputs (gripper poses, collision indicators, and states). Final action predictions are derived from the last refinement stage.

(4) RVT-2 (Goyal et al., 2024) extends RVT with a two-stage architecture: the coarse stage predicts area of interest, while the fine stage renders close-up images for precise gripper pose estimation. Beyond this multi-stage design, RVT-2 improves computational and memory efficiency through

replacing transposed convolutions with convex upsampling, optimizing network parameters, substituting PyTorch3D with a point-renderer for virtual rendering, and utilizing both global and local features to predict gripper rotation, state and collision indicator. Additionally, it reduces five static virtual viewpoints to three (front, top, right), accelerating training while maintaining performance.

(5) $\Sigma$-agent (Ma et al., 2024) integrates visual and language encoders, multi-view query transformer (MVQ-Former), contrastive imitation learning module. The visual encoder processes five virtual images with intra-view self-attention. Language instructions are encoded using CLIP and projection layers, generating language tokens for cross-attention computation. MVQ-Former transforms visual tokens into view-specific query tokens for two contrastive learning objectives: a state-language one aligns visual and text tokens in a joint embedding space to learn discriminative representations; a (state, language)-future one concatenates current visual, query and language tokens, then processes them through 4 self-attention layers to derive current-state queries. These queries are contrasted against future-state features, which are extracted by feeding next-state images to the visual encoder and applying average pooling. Both objectives augment the standard imitation learning loss during training to enhance representation learning, but are excluded during inference.

(6) SAM-E (Zhang et al., 2024) incorporates the Segment Anything Model (SAM) as a foundational visual perception module. Based on RVT's rendering strategy, it processes RGB channels through a LoRA-tuned SAM encoder, enabling generation of prompt-guided, object-oriented image embeddings. Concurrently, spatial features are extracted from depth and pixel coordinate channels via a Conv2D layer. These features are channel-wise concatenated with SAM embeddings to form composite view tokens. Combined with language tokens, these view tokens are processed by a transformer through view-wise and cross-view attention mechanisms. This generates enriched visual tokens for action-sequence prediction. Unlike step-by-step paradigms, SAM-E models coherent action sequences by enforcing temporal smoothness in end-effector poses. For translation prediction, it extends view-specific heatmaps with temporal channels. While rotation, state, and collision indicators are derived from view-fused global features following RVT.

(7) 3D-MVP (Qian et al., 2025) aims to augment visual encoder for learning generalizable representations, decomposing the view transformer into an input renderer, an encoder mapping static virtual images to latent embeddings, and an action decoder. Rather than training the RVT architecture from scratch, 3D-MVP adopts a two-stage training paradigm: first pretrains RVT encoder using masked autoencoding on large-scale 3D scene datasets, then fine-tunes it on downstream manipulation demonstrations. The finetuning procedure is identical to RVT training, while the pretraining introduces a MAE decoder to reconstruct all five virtual images from masked multi-camera tokens. This multi-view pretraining scheme produces 3D-aware features robust to occlusions and viewpoint changes, enhancing manipulation performance and robustness to environmental variations.

## C    COMPUTATIONAL EFFICIENCY AND CAPACITY ANALYSIS

**Time Efficiency.** Inference latency on an NVIDIA A800 GPU (batch size 512, averaged over 20 trials) in Fig. 4 (a) shows the dynamic-view stream consumes $\sim 1.8\times$ more time than the static-view stream (2.48 vs. $1.37 \times 10^{-3}$ min). This cost stems from the MViT backbone (Fan et al., 2021), which is computationally heavier but essential for heatmap precision (Li et al., 2018). Notably, both streams remain faster than the RVT-2 action head ($4.06 \times 10^{-3}$ min), ensuring Cortical Policy achieves superior performance (+3.5% gain over RVT-2) without compromising responsiveness. **Compute Control.** Success rate tracking (Fig. 9) shows stability after 50 epochs, confirming that performance gains stem from architectural design rather than extended training. **Parameter Capacity Control.** We further rule out the gains from mere model scaling by comparing against a "Deeper" (18-layer) RVT-2 baseline. As detailed in Table 7, Cortical Policy outperforms this heavier model (+2.6%) with fewer parameters and lower FLOPs, validating that the dual-stream mechanism, rather than model capacity, is the key driver of performance.

## D    VISUALIZATION OF GEOMETRICALLY CONSISTENT KEYPOINTS

Fig. 10 shows additional qualitative results of 3D supervision generation, demonstrating the viewpoint-consistent keypoint distributions across eight manipulation tasks. The figure organizes multi-perspective keypoint visualizations as follows:

- *Rows* (top to bottom): Close Jar, Place Cups, Sweep to Dustpan, Insert Peg, Push Buttons, Drag Stick, Screw Bulb, and Stack Blocks
- *Columns* (left to right): Coarse stage (top, front, right views) followed by fine stage (top, front, right views)

# E  VISUALIZATION OF EGOCENTRIC RENDERING

Fig. 7 compares dynamic-view options for position-aware pretraining data: raw wrist camera views versus rendered views from dynamic virtual cameras. As can be seen, end-effector positions in raw wrist camera views are fixed at constant pixel coordinates. By contrast, rendered views exhibit positional variations, with shadows indicating regions occluded from physical wrist-mounted cameras. Apart from image samples, egocentric video examples are also included in supplementary material.

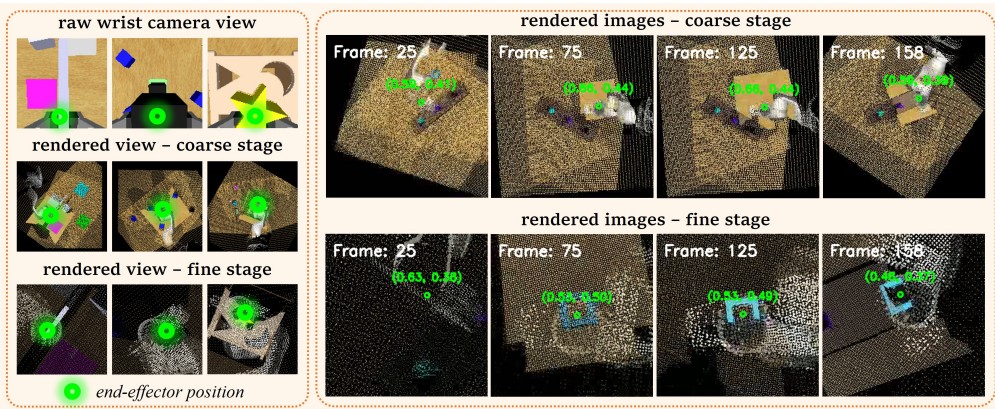

Figure 7: Comparison of dynamic egocentric views and rendered examples.

# F  ADDITIONAL FAILURE CASE ANALYSIS

This section provides a deeper analysis of the RVT-2 failure in the "stack 2 blocks in between the bottles" task (Fig. 1), investigating whether it stems from spatial reasoning deficiency, mode collapse, or language misunderstanding. As shown in Fig. 8, RVT-2 exhibits distinct action patterns between the two stacking tasks. This behavioral diversity in a novel configuration indicates both the absence of mode collapse and RVT-2's ability to adapt to the tasks with different scenes and instructions. The failure, therefore, points to a deficiency in the precise spatial reasoning required for successful placement of the "in between" relationship.

# G  DETAILED RESULTS ON COLOSSEUM

In this section, we provide comprehensive results on the COLOSSEUM benchmark (Pumacay et al., 2024), extending the analysis in Section 4.4. We evaluate the same models from Table 1 and Table 2 (all trained on the original RLBench tasks) under a suite of unseen perturbations. These perturbations encompass changes to object properties (MO/RO-Color, MO/RO-Texture, MO/RO-Size), Light Color, Table Color/Texture, Distractor, Background Texture and Camera Pose. Evaluations also include the RLBench Variations described in Table 4.

Following the official COLOSSEUM protocol for zero-shot generalization, we evaluate RVT-2, Cortical Policy, and its variants on four tasks shared by RLBench and COLOSSEUM: drag stick, place wine, stack cups, and insert peg. Results are averaged over three independent trials. For a comprehensive comparison, we include results of RVT (Goyal et al., 2023) and PerAct (Shridhar et al., 2023) from the original COLOSSEUM paper (Pumacay et al., 2024). The detailed per-task results across all perturbation types are shown in Table 6. Key observations include:

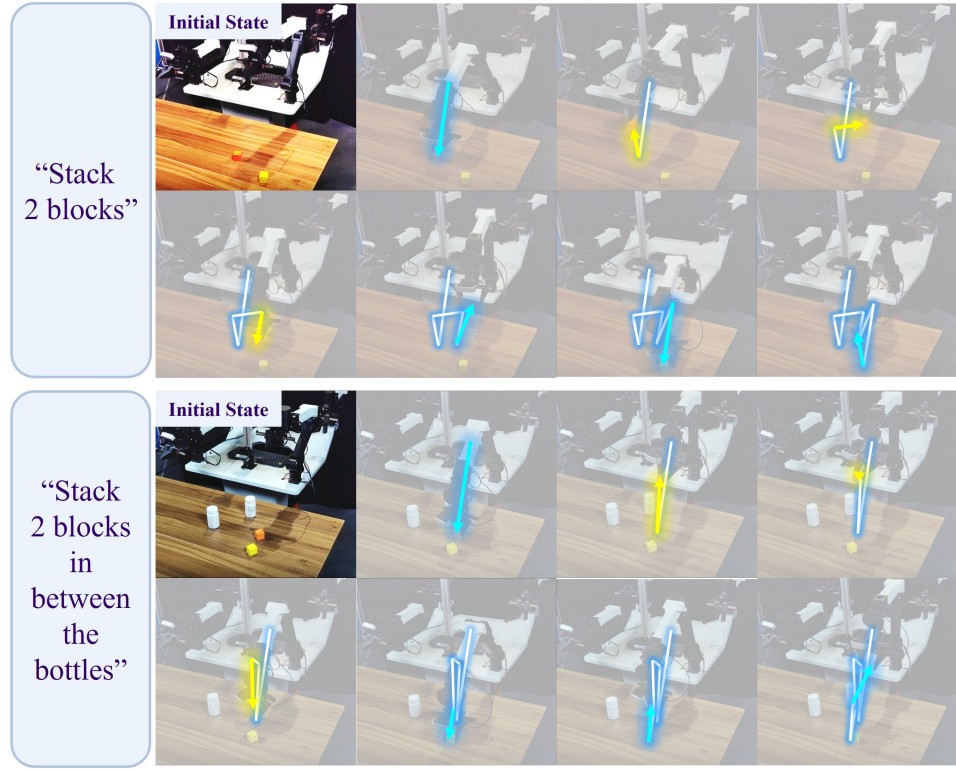

Figure 8: **RVT-2 behavior comparison.** (Top) In the basic task "stack 2 blocks", RVT-2 places the first block near the robot arm. (Bottom) In the spatial task "stack 2 blocks in between the bottles", RVT-2 attempts to place the first block near the bottles but fails due to imprecision.

- Cortical Policy achieves the highest average success rate on all four tasks (drag stick: 80.3%, place wine: 89.0%, stack cups: 76.8%, insert peg: 32.6%), demonstrating superior robustness to unseen scene configurations.
- In tasks that heavily rely on spatial reasoning, such as "stack cups", our method achieves the highest success rate (36.0%) under the combined "All Perturbations" setting, highlighting its superior robustness against geometric variations.
- Ablation results underscore the critical role of the dynamic-view stream. Specifically, variant **E** consistently outperforms **B**, with a notable margin of +9.3% in the challenging "stack cups" task.

Collectively, these COLOSSEUM results demonstrate the dual-stream architecture's effectiveness in handling realistic environmental variations, showing that the dynamic-view stream is the primary contributor to the observed generalization and robustness.

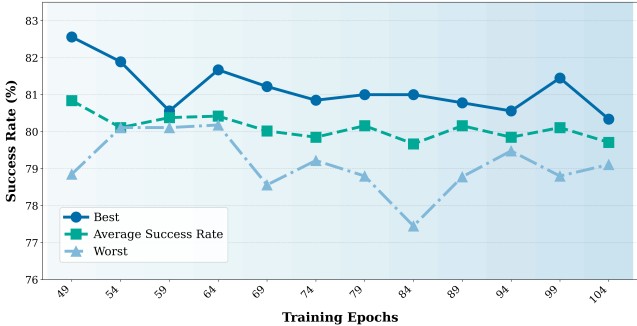

Figure 9: Success rate variations with training epochs for compute-control evaluation.

Table 6: Success rates of different methods under various perturbations of COLOSSEUM.

| Task Name | Models | Avg. Success ↑ | No Perturbations | All Perturbations | MO-Color | RO-Color | MO-Texture | RO-Texture | MO-Size |
|---|---|---|---|---|---|---|---|---|---|
| drag stick | PerAct | 17.6 | 36 | **0** | 20 | 12 | 4 | 8 | 40 |
| | RVT | 59.2 | 84 | **0** | 24 | 52 | 88 | 88 | 92 |
| | RVT-2 | 69.8 | 84.0±3.3 | 0.0±0.0 | 84.0±3.3 | 80.0±0.0 | 90.7±1.9 | 89.3±1.9 | 89.3±1.9 |
| | Variant **B** | 73.4 | 85.3±5.0 | 0.0±0.0 | 90.7±1.9 | 89.3±1.9 | 94.7±1.9 | 90.7±1.9 | 92.0±0.0 |
| | Variant **D** | 78.1 | **88.0**±0.0 | 0.0±0.0 | **92.0**±0.0 | **96.0**±0.0 | 96.0±0.0 | **92.0**±0.0 | 94.7±1.9 |
| | Variant **E** | 78.3 | **88.0**±0.0 | 0.0±0.0 | **92.0**±0.0 | **96.0**±0.0 | 97.3±1.9 | **92.0**±0.0 | 88.0±0.0 |
| | Ours | **80.3** | **88.0**±0.0 | 0.0±0.0 | **92.0**±0.0 | **96.0**±0.0 | **100.0**±0.0 | **92.0**±0.0 | **96.0**±0.0 |
| place wine | PerAct | 3.7 | 0 | 0 | 0 | 0 | – | 0 | 8 |
| | RVT | 57.4 | 60 | 12 | 72 | 40 | – | 72 | 36 |
| | RVT-2 | 84.3 | 90.7±3.8 | **44.0**±0.0 | 76.0±0.0 | 88.0±3.3 | – | 88.0±3.3 | 96.0±0.0 |
| | Variant **B** | 85.1 | 94.7±5.0 | 16.0±0.0 | 82.7±5.0 | 90.7±1.9 | – | 96.0±0.0 | 96.0±0.0 |
| | Variant **D** | 84.8 | 96.0±0.0 | 4.0±0.0 | 84.0±0.0 | 92.0±5.7 | – | 92.0±0.0 | 96.0±0.0 |
| | Variant **E** | 88.2 | 98.7±1.9 | 0.0±0.0 | 97.3±1.9 | 93.3±3.8 | – | **97.3**±3.8 | 96.0±0.0 |
| | Ours | **89.0** | **100.0**±0.0 | 0.0±0.0 | **100.0**±0.0 | **98.7**±1.9 | – | 80.0±0.0 | **100.0**±0.0 |
| stack cups | PerAct | 4 | 8 | 0 | 12 | – | 0 | – | – |
| | RVT | 13.3 | 0 | 0 | 12 | – | 12 | – | – |
| | RVT-2 | 66.3 | 96.0±0.0 | 4.0±0.0 | 76.0±0.0 | – | 96.0±0.0 | – | – |
| | Variant **B** | 67.0 | 97.3±1.9 | 5.3±1.9 | 84.0±0.0 | – | 96.0±0.0 | – | – |
| | Variant **D** | 68.0 | 88.0±0.0 | 16.0±0.0 | 80.0±0.0 | – | 92.0±0.0 | – | – |
| | Variant **E** | 76.3 | **98.7**±1.9 | 34.7±1.9 | 86.7±5.0 | – | 96.0±0.0 | – | – |
| | Ours | **76.8** | 96.0±0.0 | **36.0**±0.0 | **88.0**±0.0 | – | **100.0**±0.0 | – | – |
| insert peg | PerAct | 5.1 | 4 | 0 | 0 | 4 | – | 4 | 0 |
| | RVT | 9.1 | 4 | 0 | 0 | 16 | – | 12 | 24 |
| | RVT-2 | 21.4 | 32.0±0.0 | 12.0±0.0 | 20.0±3.3 | 26.7±1.9 | – | 21.3±1.9 | 56.0±0.0 |
| | Variant **B** | 28.4 | 36.0±0.0 | **20.0**±3.3 | 21.3±3.8 | 30.7±1.9 | – | 28.0±0.0 | **64.0**±0.0 |
| | Variant **D** | 32.0 | 36.0±0.0 | **20.0**±0.0 | 22.7±1.9 | 30.7±1.9 | – | **40.0**±3.3 | 60.0±0.0 |
| | Variant **E** | 32.1 | 38.7±1.9 | 0.0±0.0 | 24.0±0.0 | 32.0±0.0 | – | 24.0±0.0 | **64.0**±0.0 |
| | Ours | **32.6** | **42.7**±1.9 | 4.0±0.0 | **32.0**±0.0 | **36.0**±0.0 | – | 28.0±0.0 | **64.0**±0.0 |

| Task Name | Models | RO-Size | Light Color | Table Color | Table Texture | Distractor | Background Texture | RLBench Variations | Camera Pose |
|---|---|---|---|---|---|---|---|---|---|
| drag stick | PerAct | 8 | 12 | 12 | 8 | 0 | 20 | 64 | 20 |
| | RVT | 0 | 72 | 52 | 88 | 4 | **88** | 76 | 80 |
| | RVT-2 | 29.3±1.9 | 73.3±1.9 | 53.3±3.8 | 46.7±3.8 | 80.0±3.3 | 84.0±0.0 | 78.7±1.9 | 84.0±3.3 |
| | Variant **B** | 8.0±8.0 | 88.0±0.0 | 64.0±0.0 | 52.0±0.0 | 82.7±3.8 | 84.0±0.0 | 84.0±0.0 | 96.0±0.0 |
| | Variant **D** | 8.0±8.0 | 90.7±1.9 | 68.0±0.0 | 88.0±0.0 | 94.7±3.8 | 84.0±0.0 | **88.0**±0.0 | 92.0±0.0 |
| | Variant **E** | 30.7±1.9 | **92.0**±0.0 | 72.0±0.0 | 90.7±1.9 | 76.0±0.0 | 80.0±0.0 | **88.0**±0.0 | 92.0±0.0 |
| | Ours | **32.0**±0.0 | 72.0±0.0 | **76.0**±0.0 | **92.0**±0.0 | **96.0**±0.0 | 84.0±0.0 | **88.0**±0.0 | **100.0**±0.0 |
| place wine | PerAct | 12 | 8 | 0 | 4 | 0 | 4 | 8 | 8 |
| | RVT | 64 | 88 | 88 | 60 | 32 | 52 | 72 | 72 |
| | RVT-2 | 84.0±0.0 | 89.3±6.8 | 88.0±6.5 | 86.7±1.9 | 84.0±5.7 | 88.0±0.0 | 89.3±3.8 | 88.0±3.3 |
| | Variant **B** | 88.0±6.5 | 90.7±1.9 | 89.3±3.8 | 88.0±0.0 | 89.3±1.9 | 89.3±1.9 | 92.0±0.0 | 89.3±1.9 |
| | Variant **D** | 76.0±0.0 | 90.7±1.9 | 90.7±1.9 | 90.7±1.9 | 94.7±1.9 | 88.0±0.0 | 94.7±5.0 | 97.3±1.9 |
| | Variant **E** | 92.0±0.0 | 92.0±0.0 | **92.0**±0.0 | 92.0±0.0 | 96.0±3.3 | 93.3±1.9 | 96.0±3.3 | 98.7±1.9 |
| | Ours | **98.7**±1.9 | **97.3**±1.9 | 80.0±0.0 | **96.0**±3.3 | **98.7**±1.9 | **98.7**±1.9 | **97.3**±1.9 | **100.0**±0.0 |
| stack cups | PerAct | – | 0 | 16 | 0 | – | 4 | 0 | 8 |
| | RVT | – | 40 | 12 | 24 | – | 16 | 24 | 20 |
| | RVT-2 | – | 80.0±0.0 | 24.0±0.0 | 64.0±0.0 | – | 92.0±0.0 | 68.0±0.0 | 62.7±1.9 |
| | Variant **B** | – | 88.0±0.0 | 16.0±0.0 | 80.0±0.0 | – | 68.0±0.0 | 69.3±5.0 | 65.3±3.8 |
| | Variant **D** | – | 84.0±0.0 | 16.0±0.0 | 72.0±0.0 | – | 84.0±0.0 | 80.0±0.0 | 68.0±0.0 |
| | Variant **E** | – | 92.0±0.0 | **52.0**±0.0 | 72.0±0.0 | – | 72.0±0.0 | 84.0±3.3 | 74.7±1.9 |
| | Ours | – | **96.0**±0.0 | 4.0±0.0 | **84.0**±0.0 | – | **96.0**±0.0 | **92.0**±3.3 | **76.0**±0.0 |
| insert peg | PerAct | 8 | 8 | 4 | 0 | 8 | 4 | 28 | 0 |
| | RVT | 4 | 8 | 16 | 20 | 4 | 4 | 8 | 8 |
| | RVT-2 | 20.0±0.0 | 12.0±0.0 | 4.0±0.0 | 20.0±0.0 | 17.3±1.9 | 24.0±0.0 | 18.7±1.9 | 16.0±0.0 |
| | Variant **B** | 36.0±0.0 | 28.0±0.0 | 20.0±3.3 | 24.0±5.7 | 20.0±3.3 | 26.7±1.9 | 26.7±1.9 | 16.0±0.0 |
| | Variant **D** | 36.0±3.3 | 28.0±0.0 | 22.7±3.8 | **28.0**±0.0 | 48.0±0.0 | 24.0±0.0 | 36.0±0.0 | 16.0±0.0 |
| | Variant **E** | **38.7**±1.9 | **37.3**±1.9 | 24.0±0.0 | **28.0**±0.0 | 50.7±3.3 | 29.3±1.9 | 38.7±1.9 | **20.0**±0.0 |
| | Ours | 24.0±0.0 | 12.0±0.0 | **26.7**±0.0 | **28.0**±0.0 | **54.7**±1.9 | **32.0**±0.0 | **52.0**±0.0 | **20.0**±0.0 |

Table 7: Capacity-controlled ablation study on RLBench.

| Models | Parameter Size | Parameter Num | FLOPs | Avg. Success |
|---|---|---|---|---|
| RVT-2 | 277.3MB | 72.7M | 14.98 G | 77.5 |
| Baseline (Deeper) | 596.6MB | 156.4M | 25.59 G | 78.4 |
| Cortical Policy (Ours) | 551.9MB | 144.7M | 22.37 G | 81.0 |

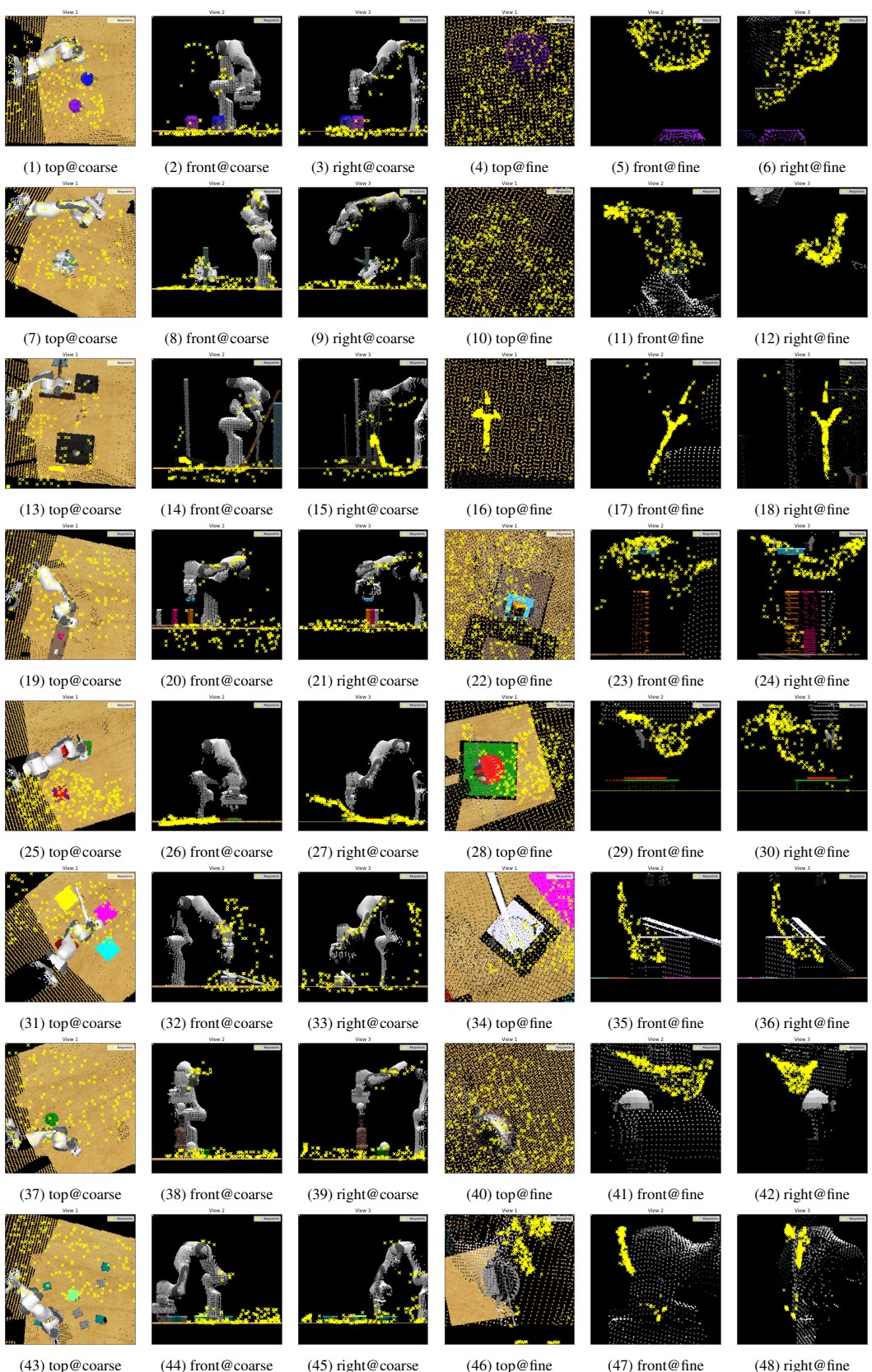

Figure 10: Additional visualization of geometrically consistent keypoints.

