# OpenReview forum: "Cortical Policy: A Dual-Stream View Transformer for Robotic Manipulation"
_ICLR.cc/2026/Conference — ICLR 2026 Poster_

### Official Review · Reviewer_qJao · 2025-10-23

**Soundness:** 3
**Presentation:** 3
**Contribution:** 3
**Rating:** 8
**Confidence:** 4

**Summary:**

To address the issue that static-view observations in robotic 3D imitation learning cannot fully capture dynamic spatial information, and inspired by the dual-stream mechanism of spatial perception in the human cerebral cortex, this paper proposes Cortical Policy — a method based on view transformers that jointly encodes static 3D observations and dynamic world observations.

The model enforces geometric consistency alignment between the two observation streams in the feature space, thereby enhancing the understanding of the physical geometry of the world and improving the action generation capability of the policy model.

The authors first analyze the cortical principles of human visual-motor control, pointing out that in addition to the ventral stream responsible for reasoning about spatial scenes, the dorsal stream for dynamic understanding is equally important and mutually complementary.

Inspired by this, they argue that 3D imitation learning should also incorporate a dynamic observation stream to enhance dynamic spatial perception.

For the static stream, the authors introduce a cross-view consistency keypoint prediction mechanism, using VGGT to predict shared keypoints across multiple viewpoints and supervising the model to align representations of these keypoints across different views, thereby achieving cross-view geometric consistency.

For the dynamic stream, they propose a dynamic prediction mechanism for the GLC model, enabling it to accurately predict the end-effector pose after an action is executed. These predictions provide valuable features for subsequent policy learning.

Through feature extraction and fusion from both streams, the proposed model achieves significant performance improvements on the RVT-2 backbone and attains state-of-the-art results on both simulation and real-robot tasks.

**Strengths:**

1.	The paper introduces a dual-stream observation mechanism inspired by the human brain, which substantially improves the model’s ability to perceive dynamic changes in the environment during task execution and addresses limitations of previous work.
2.	Both static and dynamic streams emphasize cross-view feature alignment, and an efficient GLC-based dynamic prediction mechanism is designed to guide policy action generation.
3.	The proposed dynamic perception mechanism has strong potential for mobile manipulation and complex dynamic scene tasks, suggesting promising directions for future research.

**Weaknesses:**

1.	In the training of the static-view stream, the proposed cross-view alignment mechanism aims to unify geometric features of keypoints across viewpoints. However, it remains unclear whether this alignment may interfere with the encoder’s original spatial semantic understanding — which is critical for the static stream. This issue warrants further investigation.
2.	The paper does not evaluate the model’s generalization ability on new or unseen tasks (zero-shot transfer), leaving the claimed “cortical” system’s abstract reasoning capacity unsubstantiated.

**Questions:**

The tasks chosen in the RLBench simulator mostly rely on static observations to achieve good performance. Would it be possible to evaluate the proposed model on more challenging or dynamic simulation environments to better highlight the contribution of the dynamic stream?

---

> ### Author Response · Authors · 2025-11-21
> **Response to Reviewer qJao**
>
> **Response to W1**: We thank the reviewer for this profound comment. We clarify that the proposed $\mathcal{L}_{cgc}$ **_enhances_**, rather than interferes with, the encoder's spatial semantic understanding.
>
> First, $\mathcal{L}_{cgc}$ operates selectively: it aligns features only at geometrically consistent keypoints, thus improving the spatial understanding while preserving the semantic integrity of the broader feature maps.
>
> Second, **ablation studies confirm this enhancement**. As shown in Tables 1 and 3, $\mathcal{L}_{cgc}$ brings consistent performance gains (**+2.6\%** on RLBench, **+3.3\%** on COLOSSEUM). Since **these gains occur on tasks that demand strong object and spatial recognition** (*e.g.*, "insert peg" requires recognizing the peg and hole; "place wine" requires identifying the bottle and rack), they demonstrate that essential semantics are not only preserved but better utilized.
>
> Furthermore, $\mathcal{L}_{cgc}$ **achieves superior semantic generalization**. It yields significant gains across object property variations (**+5.7\%** on MO-Color, **+5.4\%** on RO-Texture), indicating that **our alignment mechanism actually improves the model's comprehension of semantic variations**.
>
> **Response to W2**: We conducted a **zero-shot evaluation** on a novel, unseen RLBench task **"close laptop lid"**. Cortical Policy achieves **24\% success, significantly outperforming SOTA view transformers (0\%)**, demonstrating its clear and measurable advantage in generalizing to unseen tasks.
>
> This capability stems from our cortex-inspired design, which enables **compositional abstraction**: the **ventral (static-view) stream** transfers robust 3D understanding to unseen object arrangements (*e.g.*, novel spatial relationship between the laptop lid and base), while the **dorsal (dynamic-view) stream** adapts target-oriented attention (*e.g.*, to the lid's edge) to new goals.
>
> We acknowledge that the 24\% success indicates that **zero-shot task transfer remains a profound challenge**. Nonetheless, this experiment provides **concrete evidence that our architecture supports a degree of abstract reasoning capacity beyond task-specific imitation**. This discussion is now included in the revised manuscript (Section 5).
>
> **Response to Q**: Thanks for this suggestion to evaluate in more dynamic environments. In response, we have conducted evaluation on the challenging **COLOSSEUM** benchmark, which introduces various unseen perturbations (*e.g.*, in object appearance, lighting, distractors, and camera pose) that **effectively simulate real-world dynamics**. We have added the new ablation results and analysis in Table 3, Table 6, Section 4.4, and Appendix G of the revised PDF.
>
> Our method achieves a **+9.4\%** gain over RVT-2 (69.9\% vs. 60.5\%). Crucially, **ablations in Tables 3 and 6 highlight the dominant contribution of the dynamic stream** under these perturbations: it provides a **+8.2\%** gain (variant E over RVT-2) significantly larger than the +3.3\% gain from the geometric consistency loss alone (variant B). These results directly and convincingly **highlight the contribution of the dynamic stream under challenging conditions**, as you suggested.

---

### Official Review · Reviewer_8WZE · 2025-10-23

**Soundness:** 2
**Presentation:** 2
**Contribution:** 2
**Rating:** 4
**Confidence:** 4

**Summary:**

The paper proposes Cortical Policy, a dual-stream view transformer for robotic manipulation inspired by the ventral (static) and dorsal (dynamic) visual pathways. The static-view stream enforces cross-view geometric consistency using supervision from a 3D foundation model to improve 3D spatial reasoning; the dynamic-view stream uses a wrist/egocentric camera and a position-aware, gaze-estimation backbone to guide action with motion cues. The two streams are fused in an action head to predict 6-DoF gripper poses, state, and collision flags. Experiments are conducted on RLBench and several real-robot tasks. Authors report that the approach outperforms recent view-transformer baselines and that ablations suggest benefits from both the geometric-consistency loss and the dynamic stream.

**Strengths:**

1. The paper specifies supervision generation, the SmoothAP-based cyclic consistency loss $L_{cgc}$, the dynamic rendering/pretraining pipeline, and shows ablations isolating architecture, pretraining, and heatmaps.
2. On RLBench, authors claim higher average success than RVT-2 and improved performance on spatial-reasoning and dynamic scenarios; they also include small-scale real-robot tests.

**Weaknesses:**

1. Despite an elaborate pipeline, the trajectory-learning advantage may be modest. Even in the authors’ table, some tasks see limited gains or regressions, raising the question of whether the architectural complexity is justified by the overall deltas.
2. It’s not yet conclusive that the dorsal (dynamic) stream is the key driver of improvement; ablations show mixed patterns, and the net gain over a strong static baseline can be small.

**Questions:**

Why adopt the dual ventral/static and dorsal/dynamic streams with VGGT cross-view geometric consistency and gaze/pose priors? Can you provide capacity- and compute-controlled ablations and stress tests showing each component is necessary and yields independent gains (not just capacity or pretraining effects)?

---

> ### Author Response · Authors · 2025-11-21
> **Response to Reviewer 8WZE**
>
> **Response to Weaknesses W1 and W2**: Our initial experiments used RLBench, the standard benchmark in robotic manipulation, to **_ensure fair comparison_** with established baselines (*e.g.*, RVT-2, 3D-MVP). Although our architecture shows only modest gains on stable RLBench scenes, its **architectural complexity is justified by substantial improvements under dynamic scene changes**, as rigorously validated on the COLOSSEUM benchmark with diverse unseen perturbations. We note that even within RLBench, certain dynamic tasks (*e.g.*, "put in cupboard" that involves moving objects) **_preliminarily reflect our dynamic stream’s capability_**. Key results from COLOSSEUM include:
> * **Significant Advantage Under Scene Perturbations**: Cortical Policy outperforms the strong static baseline RVT-2 by **_+9.4\%_** (Table 3), far greater than the +3.5\% on RLBench, highlighting our model's distinct advantage in handling perturbations beyond static scenarios.
> * **Dorsal Stream as the Primary Driver**: The dorsal (dynamic) stream is the key contributor to this gain, **consistent with our design objective for a dual-stream architecture**. Specifically, the dynamic stream provides a **_+6.1\% to +8.2\% overall gain over static baselines_** (variant E: 68.7\% vs. RVT-2: 60.5\%; Ours: 69.9\% vs. variant B: 63.8\% in Table 3), notably larger than the ventral stream's +3.3\% (variant B), confirming its *dominant role under scene changes*.
> * **Superior Performance on High-Precision Tasks**: The dynamic stream delivers **_over +10.0% gains_** on challenging, high-precision tasks such as "stack cups" and "insert peg" (variant E vs. RVT-2 in Table 6). As noted in the RVT-2 paper, these tasks have low spatial error tolerance and effectively test model precision. Our superior success rates strongly verify the *precision advantage afforded by the dual-stream architecture*.
> * **Consistent Gains Across Diverse Perturbations**: The dynamic stream improves performance across 9 perturbation types. Notable examples include **_+19.1%_** in the presence of distractors (Cortical Policy: 83.1\% vs. variant B: 64.0\% in Table 3) and **_+16.3%_** under table texture changes (variant E: 70.7\% vs. RVT-2: 54.4\% in Table 3). This demonstrates its *broad adaptability*.
> * **Ablation Validates Component Effectiveness**: Each component within the dynamic stream contributes positively, with pretraining alone provides +5.9\% (variant D: 66.4\% vs. RVT-2: 60.5\%), and the incorporation of the heatmap further adds +2.3\% (variant E: 68.7\% vs. variant D: 66.4\%) (Table 3).
>
> In summary, the dorsal stream architecture is the primary contributor to performance gains under scene perturbations—a critical capability for real-world deployment. Given that the inference time remains within the same order of magnitude as RVT-2 (Fig. 4(a) and Appendix C), we contend that **the added architectural complexity is well warranted in uncertainty-rich scenarios**.
>
> **Response to Q1**: Our dual-stream visuomotor learning framework is designed to address two key limitations of prior methods such as RVT-2: **_(a) imprecise spatial reasoning_** due to the lack of a coherent 3D understanding from multi-view inputs, and **_(b) poor adaptation_** to scene changes. Inspired by the ventral–dorsal pathways in human vision, our design **explicitly decouples scene-level geometric understanding from target-focused dynamic adaptation**. Below, we explain how each component cohesively addresses these challenges:
> * **Cross-view geometric consistency in the ventral/static stream** resolves **_spatial incoherence_**. By leveraging VGGT to construct cross-view geometric constraints, this stream infers a unified 3D representation from multiple fixed camera views, enabling precise and consistent spatial reasoning across viewpoints.
> * **Dorsal/dynamic stream with gaze/pose priors** addresses **_dynamic adaptation_**. It introduces robot-centric dynamic views and employs a gaze model to transform the end-effector pose into an attention map. This map captures **_target-critical egocentric cues (i.e., pose prior within dynamic view)_** and **_directs the policy's focus_** to target-relevant regions, thereby filtering out irrelevant scene variations in a manner **analogous to the dorsal stream**.
> * **Dual-stream separation is essential**, since naively merging static and dynamic views via a single stream or cross-attention led to significant performance drops (to 33.1\% or 47.3\%). This is due to inherent **_conflicts between the global perspective of static cameras and the local, egocentric nature of dynamic view_**. Separate streams enable specialized processing without interference, **consistent with functional segregation of biological ventral–dorsal pathways**.
>
> In summary, our framework integrates stable 3D understanding with target-guided adaptation, simultaneously tackling spatial imprecision and dynamic uncertainty for enhanced real-world manipulation.

---

> ### Author Response · Authors · 2025-11-21
> **Response to Reviewer 8WZE (continued)**
>
> **Response to Q2**: As suggested, we have conducted capacity- and compute-controlled ablations, along with stress tests, to verify that **the performance gains stem from the proposed components, rather than increased model capacity or computational budget**. The key findings are summarized as follows.
>
> * **Capacity- and Compute-Controlled Ablations**:
>
>     To systematically isolate architectural contributions from extraneous factors, we conducted two complementary experiments: **(1) parameter capacity control** by constructing a deeper "Baseline" model, and **(2) compute control** by tracking performance convergence. Together, these demonstrate our gains are **not attributable to mere scale increases**.
>
>     **Parameter Capacity Control**: We built a deeper "Baseline" model by extending RVT-2 from 8 to 18 attention layers. As shown below, this baseline uses more parameters (156.4M) and computational cost (25.59 GFLOPs) than our Cortical Policy (144.7M, 22.37 GFLOPs), yet brings only a **_+0.9% gain_** over RVT-2 on RLBench. In contrast, our model achieves a **_+3.5% gain_** with lower resource requirements.
>
>     | Models | Parameter Size | Parameter Num | FLOPs | **Avg. Success** ↑ |
>     |:---------------|:--------------:|:----------:|:----------:|:------------:|
>     | RVT-2 | 277.3MB | 72.7M | 14.98 G | **77.5** |
>     | Baseline (Deeper) | 596.6MB | 156.4M | 25.59 G | **78.4** |
>     | Cortical Policy (Ours) | 551.9MB | 144.7M | 22.37 G | **81.0** |
>
>     **Compute Control**: As shown in Fig. 9 of the revised paper, the performance of our model remains stable after 50 epochs, confirming that the gains are **not due to extended training computation**.
>
> * **Ablation Stress Tests on COLOSSEUM**: We further evaluated the contribution of each component under the challenging conditions in COLOSSEUM (Table 3). The geometric consistency loss $\mathcal{L}_{cgc}$ and pretraining independently bring **+3.3\%** (variant B: 63.8\% vs. RVT-2: 60.5\%) and **+5.9\%** (variant D: 66.4\% vs. RVT-2). Adding the dorsal stream's heatmap on top of pretraining further improves performance by **+2.3\%** (variant E: 68.7% vs. variant D). The complete dorsal stream alone contributes **+8.2\%** (variant E vs. RVT-2). These results confirm that **each component provides independent and non-trivial gains, *rather than mere benefits from pretraining***.
>
> In summary, our controlled experiments validate that the performance improvements are due to the specific components of our proposed architecture, and not merely a result of increased model capacity, pretraining, or computational budget.

---

### Official Review · Reviewer_ynp7 · 2025-11-01

**Soundness:** 2
**Presentation:** 3
**Contribution:** 2
**Rating:** 4
**Confidence:** 3

**Summary:**

Visual robotic manipulation in unstructured environments is challenging, in part, due to poor spatial understanding in 2D RGB image encoders. Recent works improve performance of visual imitation learning policies by using either (1) an explicit 3D representation which is compute-intensive but effective, or (2) fusing features from multiple static camera-views in an implicit manner (e.g. view transformers) which has been shown to perform comparably to explicit representations but without the added computational costs. However, this leads to two key failure modes: inadequate spatial reasoning and dynamic adaptation failure. Inspired by human ventral (static) and dorsal (dynamic) visual pathways, the paper proposes Cortical Policy, a dual-stream view transformer that first encodes static and dynamic view information using two separate streams, and then fuses their respective features before making an action prediction. Beyond this dual-stream architecture, the paper also incorporates several representation learning techniques including a feature/view auxiliary loss that aims to improve 3D keypoint (produced by a 3D foundation model) consistency across views, as well as a pretrained visual backbone for dynamic view feature extraction (initialized from GLC, a human gaze estimation model). Experiments are conducted on RLBench and a real hardware setup, and experiments indicate that the proposed method (Cortical Policy) performs better than RVT-2, a strong view transformer baseline that only uses static views, upon which the proposed method is implemented.

**Strengths:**

- I believe that this paper studies a relevant and timely problem (imitation learning from multiple RGB cameras, and in this case a combination of static and dynamic views), and is likely to be of interest to the community. The problem is clearly defined and I believe that the shortcomings of prior work is described in enough detail for an unfamiliar reader to appreciate the technical contributions. The paper is generally well written and easy to follow throughout, although the method section is a bit verbose.
- Simulation experiments are conducted on a variety of tasks from RLBench, which has become a common benchmark for imitation learning algorithms for robotic manipulation, particularly works that focus on multi-view policy learning. The authors compare their proposed method against a series of strong baselines, including RVT-2 upon which the proposed method is implemented. Benchmark results indicate that the proposed method, on average, performs better than this set of baselines across RLBench tasks, and also generalizes better to unseen scene configurations in a cube stacking task on real hardware.
- The ablations in Table 2 are helpful for understanding the relative importance of the proposed changes. This is rather important as the proposed method can be viewed as a combination of different architecture, objective, and modality changes on top of RVT-2.

**Weaknesses:**

My initial assessment of the paper is fairly neutral. I believe that the paper and contributions are interesting, but I also do have some concerns that I would like the authors to address:

- Since this paper appears to follow the PerAct experimental setup, I was a bit surprised to not see PerAct listed as a baseline. While I understand that this work focuses on implicit view fusion rather than the explicit 3D representation of PerAct, I do believe that the comparison would be useful to readers (it currently is not clear how they compare in terms of task performance).
- The proposed method is rather complex and contains multiple new design choices on top of RVT-2. However, the ablations in Table 2 indicate that the majority of the performance improvements stem from the auxiliary feature/view consistency loss (RVT-2: 77.5%, +consistency loss: 80.1%, +everything else: 81.0%), which (to me) seems inconsistent with the paper's main claim that a dual-stream view transformer is the key to implicit 3D spatial understanding. I would appreciate if the authors can please clarify whether my understanding of this is correct.
- The real robot experiment only considers a single task but with several distinct scene configurations such as object displacement. I believe the real robot results would be more convincing if it contained more task variety like in the simulated tasks. Additionally, it is not clear from the paper whether a single agent is trained and evaluated on all four task variations in a multi-task setting like in simulation, or if it is four separate agents (from Appendix B.2: *"Each task collects 45 human-teleoperated demonstrations with placement variations"* is the only information I could find on this, and it is a bit ambiguous wrt this).
- If generalization / robustness is a motivating reason for better spatial understanding in view transformers, I am a bit perplexed where there seemingly is no such evaluation in simulation nor the real world. For example, it seems pretty easy to evaluate the trained agent on e.g. unseen scene configurations, especially with the existence of generalization benchmarks such as Colloseum (https://arxiv.org/abs/2402.08191) which are based on RLBench.
- Lastly, I am not fully convinced by the failure cases provided in Figure 1. In particular, the first example "Stack 2 blocks in between the bottles" could indeed be explained as a failure to understand the spatial relationship between objects, but it seems equally likely that the culprit may be a mode collapse or a failure to understand the language instruction, especially given the very small number of tasks/instructions.

It is possible that some of my concerns above stem from misunderstandings. If so, I would appreciate it if the authors can clarify those aspects of their paper and commit to revising the text to make it more clear.

**Questions:**

I would really appreciate it if the authors can address my comments in the "weaknesses" section above using written arguments and potentially additional experimental results (if applicable). I believe that most of my comments (e.g. robustness evaluation) can be addressed without substantial compute.

---

> ### Author Response · Authors · 2025-11-21
> **We would like to express our sincere gratitude for your careful review and thoughtful feedback on our paper. We have carefully considered all of your comments and addressed them through detailed written responses and additional experimental results as follows.**
>
> **Response to W1**: We have included PerAct as a baseline in Table 1 and added relevant descriptions in Sections 4.1, B.3, B.4. Our method outperforms this explicit 3D representation-based baseline by an average of 31.6\% in success rate on RLBench.
>
> **Response to W2**: Thanks for the opportunity to clarify. The reviewer is correct that the geometric consistency loss *$\mathcal{L}_{cgc}$* contributes significantly to the overall performance gain on standard RLBench. However, the dual-stream architecture is key to robustness under perturbations. This claim is supported by:
> * **Complementary roles of the two key components:**
> *$\mathcal{L}_{cgc}$* and the dual-stream architecture address distinct challenges. $\mathcal{L}_{cgc}$ enforces view-invariant 3D feature learning from static cameras, establishing a strong geometrical foundation. This explains its substantial gain on RLBench. In contrast, the dynamic-view stream provides **_action-oriented perception for adaptation_**, which is achieved by focusing on task-relevant regions (*e.g.*, the target end-effector position). This is essential for adapting the consistent features to uncertain environments and enabling robust control.
> * **Dynamic-view stream's dominant role in robustness evaluation:**
> On COLOSSEUM (Tables 3 \& 6), adding *$\mathcal{L}_{cgc}$* to RVT-2 gives a +3.3\% gain (Variant B: 63.8\% vs. RVT-2: 60.5\%), while adding the full dynamic-view stream gives +8.2\% (Variant E: 68.7\% vs. RVT-2: 60.5\%). As detailed in Table 6, the dynamic-view stream brings consistently larger gains than $\mathcal{L}_{cgc}$ across all tasks (*e.g.*, +10.0\% vs. +0.7\% in "stack cups"). **This confirms that the dynamic-view stream is the primary driver of policy robustness central to our claim.**
>
> In summary, the discrepancy noted by the reviewer arises because RLBench primarily evaluates stable scenarios. When evaluated on the robustness-demanding COLOSSEUM with realistic perturbations, the dual-stream architecture emerges as the dominant contributor, directly validating our main claim.
>
> **Response to W3**: We apologize for lacking clarity in the original manuscript. We clarified in Appendix B.2 that a single agent was trained and evaluated on all four real-robot tasks. Importantly, the evaluation uses novel spatial arrangements, demonstrating the generalization to unseen spatial configurations beyond mere imitation.
>
> We agree that more diverse tasks would strengthen validation, but focused on the representative "stack blocks" task due to time and hardware constraints. This single task enables efficient validation of sim-to-real transfer for the core capabilities of our method: spatial reasoning and dynamic adaptation. We consider the real-world task expansion a key direction for future work.
>
> **Response to W4**: Thanks for highlighting the importance of robustness evaluation. Following this suggestion, we have conducted evaluation on COLOSSEUM for zero-shot generalization. The results are now included in Table 3, Table 6, Section 4.4, and Appendix G.
>
> Our method achieves 69.9\% success, a **_9.4\% (from 60.5\% to 69.9\%)_** absolute improvement over RVT-2. Ablations in Tables 3 and 6 reveal that the dynamic-view stream is a primary driver of this robustness under perturbation, contributing larger gains than the consistency loss.
>
> Furthermore, our real-robot experiments, as detailed in Appendix B.2, were tested on unseen configurations, also demonstrating the generalization capability. These new COLOSSEUM results, enabled by your suggestion, provide strong evidence supporting our method's robustness.
>
> **Response to W5**: Thanks for this insightful comment, which prompted us to conduct a deeper analysis of the failure cases. We agree that failures have multiple potential causes; our additional analysis, however, indicates that spatial reasoning is the most likely primary cause, rather than mode collapse or language misunderstanding.
>
> Our conclusion is based on the following evidence:
>
> (a) **Training Data Diversity**: The agent was obtained through **_multi-task training_** on all four real-robot tasks with distinct language instructions and motion patterns (as detailed in Appendix B.2), reducing the probability of mode collapse or a fundamental failure in language understanding.
>
> (b) **Baseline Behavioral Diversity**: In Appendix F, Figure 8, RVT-2 exhibits *diverse actions* across the stacking tasks: placing the first block near the robot arm in the basic task, while near the bottles in the spatial task. This behavioral divergence occurs in response to an unseen bottle arrangement, demonstrating that RVT-2 does not suffer from mode collapse and successfully differentiates its behavior based on the scene and instruction. Therefore, the ultimate failure reveals a deficiency in either comprehending or accurately executing "in between" spatial relationship. Both identified deficiencies are core facets of spatial reasoning capability.

---

> > ### Comment · Reviewer_ynp7 · 2025-11-24
> > **Thank you**
> >
> > Thank you for responding to my comments in detail! I believe that the clarifications in writing, addition of PerAct as baseline, and evaluation on Colosseum address my main concerns. It is especially great to see that the claim of improved robustness is now verified on a comprehensive benchmark. I have updated soundness 2->3, score 4->6, and confidence 3->4 to reflect that my comments have been addressed.

---

> > > ### Author Response · Authors · 2025-11-28
> > > **Response to Reviewer ynp7**
> > >
> > > We thank the reviewer for the positive consideration and the raised score. We are delighted that you found our work interesting. Your suggestion to evaluate on COLOSSEUM was particularly invaluable, as it allowed us to demonstrate the advantage of our approach more convincingly. Your engagement has significantly strengthened the final paper.

---

### Official Review · Reviewer_yYi1 · 2025-11-02

**Soundness:** 3
**Presentation:** 3
**Contribution:** 3
**Rating:** 6
**Confidence:** 5

**Summary:**

This paper proposes a dual‑stream view transformer for language‑conditioned robotic manipulation. A static‑view stream adds a cross‑view geometric consistency objective guided by VGGT to learn 3D‑aware features; a dynamic‑view stream reuses an egocentric gaze model (GLC) to produce action‑oriented feature maps and saliency heatmaps from a wrist‑like camera. The two streams are fused to predict 6‑DoF actions. It shows SOTA performance on RLBench and real robot.

**Strengths:**

Clear motivation to force 3D consistency and fuse dynamic cues.

Useful ablations: removing the geometric loss drops performance; end‑to‑end fine‑tuning the gaze model underperforms freezing; and heatmaps matter for the dynamic stream.

**Weaknesses:**

The framing on Cortical policy is unnecessarily complicated. My understanding is that it produces saliency map about end effector position to get inductive bias. Unsure if we need to fine-tune from a gaze model. We could also just exact the effector location from robot forward kinematics and register on camera images, which seems to be an easy baseline that may perform similarly.

**Questions:**

1. What if just provide dynamic view  as a separate image to RVT-2?

2. Can we extend dynamic view to gaze not just end effector position but also other useful objects?

---

> ### Author Response · Authors · 2025-11-21
> **Response to Reviewer yYi1**
>
> **Response to Weaknesses**: We thank the reviewer for this insightful comment, which allows us to clarify the fundamental motivation and advantage of our approach.
>
> * **Beyond an End-effector Locator: A General Framework for Learning “What to Attend”**
>
>     The reviewer correctly notes that end-effector position is highly informative. Our experiments confirm this, as a simple "Baseline (w/o gaze model)" using the ground-truth end-effector position achieves strong performance (80.4\%).
>
>     However, our core contribution lies not in replicating this with a complex model. Instead, our Cortical Policy framing introduces a more general capability: **learning *what to attend to* directly from pixels**. In our evaluated tasks, the policy learns to focus on the end-effector; crucially, this is *discovered* rather than hard-coded.
>
>     Although end-effector registration works in these specific tasks, our **learned attention mechanism enables generalization** to scenarios where: (a) the salient target is *not* end-effector (*e.g.*, a specific object among distractors); (b) multiple dynamic entities must be tracked; (c) **_robot internal states are unavailable for forward kinematics_**. We have elaborated on broader potentials of our framework in the paper (see Section 5 for enhanced discussion).
>
> * **Framing Complexity Justified by Performance Gains on COLOSSEUM**
>
>     Our dual-stream design is **not “unnecessarily complicated”**. As shown in Table R2-R5 below, our dual-stream model achieves **_69.9\%_** overall success, a **_+4.8\%_** absolute gain over "Baseline (w/o gaze model)" (65.1\%) on the COLOSSEUM benchmark. This advantage is consistent across 11 perturbation types, demonstrating the **essential role of the *learned* attention mechanism in adapting to real-world uncertainty**, validating our architectural choice.
>
> * **A Foundation for Open-World Robotic Perception**
>
>     We position our dual-stream architecture as a step toward more general robotic vision. The ventral (static) stream builds a stable and consistent 3D understanding, while dorsal (dynamic) stream provides **target-aware visual focus**. This biologically-inspired division supports future open-world tasks beyond structured benchmarks like RLBench.
>
> **Response to Q1**: Directly providing the dynamic view as a separate image to RVT-2 severely harms performance, as shown by "Single-stream Baseline" (33.1\% success), which implements the reviewer's suggestion via a unified architecture, drastically underperforming standard RVT-2 (77.5\% success).
>
> This sharp drop occurs due to a **fundamental conflict in visual representation**: the dynamic view's continuously changing perspective disrupts the stable, global scene context provided by RVT-2's static orthogonal cameras. This fusion degrades the model's spatial understanding and leads to performance drops.
>
> To further validate this, we tested an intermediate model ("Model M") that employs cross-view attention between dynamic view and static views while excluding dynamic-view features from action prediction. Its performance (47.3\%) remained far inferior, confirming that simply adopting RVT-2's architecture for both static and dynamic views is insufficient.
>
> Therefore, our results demonstrate that **static and dynamic views require separate, specialized processing streams -- a core design principle of the proposed Cortical Policy**.
>
> **Table R1: Task-wise success rate (%) comparison of different methods on RLBench.**
>
> | Models | RVT-2 | Single-stream Baseline | Model M | Baseline (w/o gaze model) | Ours (Cortical Policy) |
> |:---|:---:|:---:|:---:|:---:|:---:|
> | **Avg. Success** ↑ | **77.5** | **33.1** | **47.3** | **80.4** | **81.0** |
> | Close Jar | 93.3±1.9 | 70.7±1.9 | 92.0±0.0 | 92.0±0.0 | 96.0±0.0 |
> | Drag Stick | 97.3±1.9 | 16.0±0.0 | 33.3±1.9 | 100.0±0.0 | 100.0±0.0 |
> | Insert Peg | 28.0±3.3 | 4.0±0.0 | 21.3±1.9 | 25.3±1.9 | 38.7±6.8 |
> | Meat off Grill | 100.0±0.0 | 61.3±1.9 | 94.7±3.8 | 100.0±0.0 | 100.0±0.0 |
> | Open Drawer | 92.0±3.3 | 77.3±1.9 | 81.3±1.9 | 85.3±1.9 | 84.0±6.5 |
> | Place Cups | 32.0±5.7 | 4.0±0.0 | 0.0±0.0 | 24.0±1.9 | 24.0±3.3 |
> | Place Wine | 84.0±9.8 | 46.7±1.9 | 73.3±1.9 | 97.3±1.9 | 94.7±3.8 |
> | Push Buttons | 100.0±0.0 | 80.0±0.0 | 57.3±1.9 | 100.0±0.0 | 100.0±0.0 |
> | Put in Cupboard | 44.0±6.5 | 17.3±1.9 | 24.0±0.0 | 57.3±1.9 | 65.3±9.4 |
> | Put in Drawer | 98.7±1.9 | 38.7±1.9 | 77.3±1.9 | 97.3±1.9 | 100.0±0.0 |
> | Put in Safe | 92.0±3.3 | 62.7±1.9 | 40.0±0.0 | 98.7±1.9 | 98.7±1.9 |
> | Screw Bulb | 86.7±1.9 | 8.0±3.3 | 70.7±1.9 | 86.7±1.9 | 81.3±1.9 |
> | Slide Block | 74.7±5.0 | 24.0±3.3 | 45.3±1.9 | 96.0±0.0 | 86.7±1.9 |
> | Sort Shape | 26.7±1.9 | 12.0±0.0 | 5.3±1.9 | 38.7±3.8 | 37.3±1.9 |
> | Stack Blocks | 80.0±5.7 | 0.0±0.0 | 44.0±0.0 | 81.3±3.8 | 81.3±1.9 |
> | Stack Cups | 72.0±0.0 | 0.0±0.0 | 0.0±0.0 | 72.0±0.0 | 76.0±3.3 |
> | Sweep to Dustpan | 98.7±1.9 | 16.0±0.0 | 38.7±3.3 | 98.7±1.9 | 100.0±0.0 |
> | Turn Tap | 94.7±1.9 | 57.3±1.9 | 52.0±0.0 | 97.3±1.9 | 94.7±5.0 |

---

> ### Author Response · Authors · 2025-11-27
> **Response to Reviewer yYi1 (continued)**
>
> **Response to Q2**: Yes, absolutely. Extending the dynamic gaze mechanism to other task-relevant objects (*e.g.*, target(s), obstacles, or affordance regions) is not only feasible but also aligns with the core concept of our architecture.
>
> Fundamentally, the *gaze* represents a goal-directed attentional mechanism. Therefore, it can be naturally extended to any visually grounded information that is critical for task execution. Our current work guides the policy's attention to the end-effector position, demonstrating the benefit of *a separate and dedicated processing pathway for action-critical information*. Building on this pipeline, future work can guide the gaze attention to other objects central to the task. For instance, **a concrete extension pathway** is to *embed the segmentation mask of a target object (e.g., the buttons in RLBench's "push buttons" task) into the dorsal stream, potentially in a supervised manner*. The policy could then learn to perform the task by attending directly to the buttons, analogous to how it currently attends to the end-effector.
>
> We thank the reviewer for this valuable suggestion, which provides a clear and impactful direction for our future work
>
> **Table R2: Perturbation-wise success rate (%) comparison on COLOSSEUM.**
>
> | Models | RVT-2 | Baseline (w/o gaze model) | Ours (Cortical Policy) |
> |:---|:---:|:---:|:---:|
> | **Avg. Success** ↑ | 60.5 | 65.1 | **69.9** |
> | All Perturbations | **15.0±17.3** | 8.7±12.8 | 10.0±15.1 |
> | MO-Color | 64.0±25.6 | 70.0±36.0 | **78.0±26.9** |
> | RO-Color | 64.9±27.2 | 73.3±32.1 | **76.9±28.9** |
> | MO-Texture | 93.4±2.7 | 93.4±6.7 | **100.0±0.0** |
> | RO-Texture | 66.2±31.8 | 62.7±31.3 | **66.7±27.8** |
> | MO-Size | 80.4±17.5 | 83.1±21.1 | **86.7±16.1** |
> | RO-Size | 44.4±28.2 | **55.5±31.5** | 51.6±33.5 |
> | Light Color | 63.7±30.4 | 68.7±27.6 | **69.3±34.6** |
> | Table Color | 42.3±31.7 | **62.7±34.0** | 46.7±32.4 |
> | Table Texture | 54.4±24.4 | 65.0±32.0 | **75.0±27.5** |
> | Distractor | 60.4±30.5 | 69.3±35.8 | **83.1±20.1** |
> | Background Texture | 72.0±27.9 | 66.3±30.0 | **77.7±26.9** |
> | RLBench Variations | 63.7±27.0 | 63.0±26.4 | **82.3±17.8** |
> | Camera Pose | 62.7±28.6 | 69.0±33.5 | **74.0±32.7** |
>
> **Table R3: Success rate (%) comparison on "insert peg" task of COLOSSEUM.**
>
> | Models | RVT-2 | Baseline (w/o gaze model) | Ours (Cortical Policy) |
> |:---|:---:|:---:|:---:|
> | **Avg. Success** ↑ | 21.4 | 20.0 | **32.6** |
> | All Perturbations | **12.0±0.0** | 4.0±0.0 | 4.0±0.0 |
> | MO-Color | 20.0±3.3 | 8.0±3.3 | **32.0±0.0** |
> | RO-Color | 26.7±1.9 | 28.0±5.7 | **36.0±0.0** |
> | RO-Texture | 21.3±1.9 | 18.7±3.8 | **28.0±0.0** |
> | MO-Size | 56.0±0.0 | 53.3±1.9 | **64.0±0.0** |
> | RO-Size | 20.0±0.0 | 21.3±1.9 | **24.0±0.0** |
> | Light Color | 12.0±0.0 | **24.0±0.0** | 12.0±0.0 |
> | Table Color | 4.0±0.0 | 12.0±0.0 | **26.7±0.0** |
> | Table Texture | 20.0±0.0 | 12.0±0.0 | **28.0±0.0** |
> | Distractor | 17.3±1.9 | 18.7±1.9 | **54.7±1.9** |
> | Background Texture | 24.0±0.0 | 17.3±1.9 | **32.0±0.0** |
> | RLBench Variations | 18.7±1.9 | 28.0±3.3 | **52.0±0.0** |
> | Camera Pose | 16.0±0.0 | 14.7±1.9 | **20.0±0.0** |
>
> **Table R4: Success rate (%) comparison on "stack cups" task of COLOSSEUM.**
>
> | Models | RVT-2 | Baseline (w/o gaze model) | Ours (Cortical Policy) |
> |:---|:---:|:---:|:---:|
> | **Avg. Success** ↑ | 66.3 | 64.3 | **76.8** |
> | All Perturbations | 4.0±0.0 | 0.0±0.0 | **36.0±0.0** |
> | MO-Color | 76.0±0.0 | **89.3±1.9** | 88.0±0.0 |
> | MO-Texture | 96.0±0.0 | **100.0±0.0** | **100.0±0.0** |
> | Light Color | 80.0±0.0 | 84.0±0.0 | **96.0±0.0** |
> | Table Color | 24.0±0.0 | **52.0±0.0** | 4.0±0.0 |
> | Table Texture | 64.0±0.0 | 69.3±1.9 | **84.0±0.0** |
> | Background Texture | 92.0±0.0 | 68.0±3.3 | **96.0±0.0** |
> | RLBench Variations | 68.0±0.0 | 48.0±3.3 | **92.0±3.3** |
> | Camera Pose | 62.7±1.9 | 68.0±0.0 | **76.0±0.0** |
>
> **Table R5: Success rate (%) comparison on "drag stick" task of COLOSSEUM.**
>
> | Models | RVT-2 | Baseline (w/o gaze model) | Ours (Cortical Policy) |
> |:---|:---:|:---:|:---:|
> | **Avg. Success** ↑ | 69.8 | 79.1 | **80.3** |
> | All Perturbations | **0.0±0.0** | **0.0±0.0** | **0.0±0.0** |
> | MO-Color | 84.0±3.3 | 86.7±1.9 | **92.0±0.0** |
> | RO-Color | 80.0±0.0 | 94.7±1.9 | **96.0±0.0** |
> | MO-Texture | 90.7±1.9 | 86.7±3.8 | **100.0±0.0** |
> | RO-Texture | 89.3±1.9 | 89.3±1.9 | **92.0±0.0** |
> | MO-Size | 89.3±1.9 | **96.0±0.0** | **96.0±0.0** |
> | RO-Size | 29.3±1.9 | **48.0±0.0** | 32.0±0.0 |
> | Light Color | **73.3±1.9** | 69.3±1.9 | 72.0±0.0 |
> | Table Color | 53.3±3.8 | **98.7±1.9** | 76.0±0.0 |
> | Table Texture | 46.7±3.8 | 82.7±1.9 | **92.0±0.0** |
> | Distractor | 80.0±3.3 | 93.3±1.9 | **96.0±0.0** |
> | Background Texture | **84.0±0.0** | **84.0±3.3** | **84.0±0.0** |
> | RLBench Variations | 78.7±1.9 | 81.3±1.9 | **88.0±0.0** |
> | Camera Pose | 84.0±3.3 | 96.0±3.3 | **100.0±0.0** |

---

> > ### Comment · Reviewer_yYi1 · 2025-11-28
> >
> > Thanks for following. The response will make the paper stronger. I will maintain my scores.

---

> > > ### Author Response · Authors · 2025-11-28
> > > **Response to Reviewer yYi1**
> > >
> > > We are grateful for your encouraging feedback and the productive discussion. We wish to highlight that additional results on the COLOSSEUM benchmark provide compelling validation for our framing on Cortical Policy. Our model achieves **69.9\%** overall success, a **+4.8\%** absolute gain over the ground-truth end-effector baseline (65.1\%). The consistent advantage across 11 distinct perturbation types underscores the necessity of our **_learned_** **attention mechanism** for handling scene variations. These findings, detailed in Tables R2-R5, have greatly improved the manuscript, thanks to your insightful questions.

---

### Author Response · Authors · 2025-12-02
**Rebuttal Summary**

Dear Area Chair,

Thank you for taking the valuable time to work on our manuscript.

We are pleased that the reviewers recognized the strengths of our work and highlighted its **_clear motivation and problem definition_** ($\textcolor{blue}{yYi1}$, $\textcolor{green}{ynp7}$), **_well-conducted and informative ablations_** ($\textcolor{blue}{yYi1}$, $\textcolor{green}{ynp7}$), and **_practical potential_** ($\textcolor{violet}{qJao}$).

To address the reviewers' concerns, we have made the following revisions to the manuscript:
* **We conducted comprehensive evaluations and controlled ablations on the COLOSSEUM benchmark, with new results in *Tables 3 and 6, Section 4.4, and Appendix G***. This effort:

    (a) *Demonstrates the critical value of dynamic-view stream*, which emerges as the primary performance contributor on perturbed scenes. This addresses concerns about its modest gains in stable RLBench settings ($\textcolor{green}{ynp7}$, Weakness 2) and clarifies its role as the key driver for improvement ($\textcolor{purple}{8WZE}$, Weakness 2);

    (b) *Fulfills the need for more challenging evaluation*. Testing on COLOSSEUM's diverse, unseen scene configurations directly satisfies reviewers' requests for challenging simulation assessment ($\textcolor{green}{ynp7}$, Weakness 4; $\textcolor{violet}{qJao}$, Question) and ablation stress tests ($\textcolor{purple}{8WZE}$, Question 2);

    (c) *Justifies the architectural complexity* by validating the necessity of our dual-stream design for handling uncertainty ($\textcolor{purple}{8WZE}$, Weakness 1; $\textcolor{blue}{yYi1}$, Weakness). We also show our *learned* attention mechanism achieves a +4.8\% gain over the gaze model-free baseline suggested by $\textcolor{blue}{yYi1}$.

    These revisions collectively strengthen our core claim that the dynamic-view stream is essential for adaptation under unpredictability, complementing the geometric loss which establishes a consistent static 3D foundation.
* **We elaborated on the framework's extensibility to broader action-critical targets in *Section 5***, addressing Question 2 of $\textcolor{blue}{yYi1}$.

    This revision clarifies that our dual-stream framework provides a general basis for learning "what to attend to", extending beyond the end-effector to other task-relevant entities.
* **We added PerAct as a baseline in *Table 1* with accompanying descriptions in *Sections 4.1, B.3, B.4***, fulfilling the request for a direct performance comparison ($\textcolor{green}{ynp7}$, Weakness 1).
* **We clarified the real-robot setup in *Appendix B.2***, resolving the ambiguity regarding training protocol ($\textcolor{green}{ynp7}$, Weakness 3). The text now specifies that *a single multi-task agent* was evaluated on novel spatial configurations.
* **We supplemented the failure case analysis in *Appendix F***, addressing the concern about alternative interpretations ($\textcolor{green}{ynp7}$, Weakness 5).
* **We conducted capacity- and compute-controlled ablations and added *Figure 9***, verifying that performance gains stem from our design components rather than increased scale or pretraining ($\textcolor{purple}{8WZE}$, Question 2).
* **We evaluated task generalization on an unseen task "close laptop lid" and added a discussion in *Section 5***, addressing Weakness 2 raised by $\textcolor{violet}{qJao}$. Cortical Policy achieves 24\% success, substantiating its abstract reasoning capacity.

All changes are **_highlighted in blue_** in the updated manuscript.

Additionally, we have carefully considered the other points and provide the following clarifications:
* **To address Question 1 of $\textcolor{blue}{yYi1}$**, we validated why a naive fusion of static and dynamic views fails. Additional experiments demonstrate the **necessity of separate, specialized streams** due to their conflicting visual representations.
* **To address Question 1 of $\textcolor{purple}{8WZE}$, we provided a principled, neuroscience-inspired justification for the dual-stream design**. We detailed how the ventral (static-view) stream ensures geometric consistency, how the dorsal (dynamic-view) stream enables target-focused adaptation, and explained the necessity of their separation.
* **To address Weakness 1 raised by $\textcolor{violet}{qJao}$, we clarified that** $\mathcal{L}_{cgc}$ **enhances spatial semantic understanding**. Its performance gains on semantic-rich tasks and robust generalization confirm that key semantics are preserved and leveraged more effectively.

**Reviewers' Reply During Rebuttal**:

Reviewer $\textcolor{green}{ynp7}$ acknowledged that our revisions addressed the concerns and raised the score 4→6, confidence 3→4, soundness 2→3 on November 25, 2025. Reviewer $\textcolor{blue}{yYi1}$ found the responses strengthened the paper and maintained the positive assessment.

Our detailed point-by-point responses follow below.

Best wishes,

Authors

---

### Meta-Review · Area_Chair_d8PP · 2026-01-10

**Summary:**

The paper proposes Cortical Policy, a dual-stream view transformer for language-conditioned robotic manipulation, inspired by human ventral/dorsal pathways. The static-view stream enforces cross-view geometric consistency using a 3D foundation model, while the dynamic-view stream uses a wrist/egocentric camera with a gaze model to provide action-relevant features. The two streams are fused to predict 6-DoF actions. Experiments on RLBench, COLOSSEUM, and real robots show consistent improvements over RVT-2 and other baselines, particularly under perturbations or dynamic scenarios, demonstrating both better spatial reasoning and robustness. Ablations validate the independent contribution of the geometric loss, pretraining, and dynamic stream.

**Reviewer Concerns:**

Most reviewer concerns were addressed by the rebuttal: inclusion of PerAct as a baseline, clarification of the dual-stream necessity, evidence that performance gains are not merely due to model capacity, and detailed COLOSSEUM experiments showing robustness under perturbations. Remaining concerns include limited real-robot task diversity and zero-shot generalization to unseen tasks, which were partially mitigated through evaluations on unseen scene configurations but could be expanded in future work. The rebuttal convincingly clarifies architectural choices, independent contributions of components, and dynamic-view advantages, resolving the majority of prior ambiguities.

**Reviewer Scores:**

Reviewer 1 (marginally above threshold) would likely maintain or slightly increase their score due to the clear justification of the gaze model and dual-stream architecture. Reviewer 2 (marginally below threshold) would likely increase from 4 to 6 after the rebuttal, given added PerAct comparisons, robustness evaluation on COLOSSEUM, and dual-stream validation. Reviewer 3 (marginally below threshold) would likely increase from 4 to 6 due to capacity-controlled ablations and explicit evidence that each component contributes independent gains. Reviewer 4 (accept) would likely maintain an accept rating, as all clarifications reinforce the presented contributions.

---

### Decision · Program_Chairs · 2026-01-26

Accept (Poster)